# Memory as a Markov Matrix:
# Sample Efficient Knowledge Expansion via Token-to-Dictionary Mapping

**Kaustubh Pethkar** [1]   **Ziyang Xiong** [2]   **Zuofeng Shang** [1]   **Yingcong Li** [1]

## Abstract

Continual incorporation of new knowledge is essential for the long-term evolution of large language models (LLMs). Existing approaches typically rely on parameter-update algorithms to mitigate catastrophic forgetting, yet they suffer from fundamental limitations: 1) forgetting is unavoidable as the amount of newly injected knowledge grows; and 2) model updates are often irreversible. As modern LLMs become increasingly expressive, it is natural to question whether large-scale weight updates are necessary for acquiring a small amount of new knowledge. In this work, we propose a principled framework that models autoregressive language generation as a Markov process over tokens, where model memory is represented by a Markov transition matrix. Under this formulation, incorporating new knowledge/tokens corresponds to extending the state space, and preserving existing transitions guarantees retention of previously learned knowledge. We then prove a sample complexity bound for incorporating new tokens via a token-to-dictionary mapping strategy. In particular, for learning the transition behavior of each new token, the required number of samples scales linearly with the number of existing tokens it is mapped to. To realize this mapping, we propose an embedding-tuning algorithm that requires minimal parameter updates and induces zero forgetting. Experimental results further demonstrate the effectiveness of our method and validate our theoretical findings.

[1]{kp759,zuofeng.shang,yingcong.li}@njit.edu, New Jersey Institute of Technology. [2]xziyang@berkeley.edu, University of California, Berkeley. Correspondence to: Yingcong Li <yingcong.li@njit.edu>.

*Proceedings of the 43rd International Conference on Machine Learning*, Seoul, South Korea. PMLR 306, 2026. Copyright 2026 by the author(s).

## 1. Introduction

Large language models (LLMs) acquire extensive linguistic and factual knowledge during pretraining, yet adapting them to new vocabulary, entities, or domains remains challenging (Lewis et al., 2020; Muennighoff et al., 2023; Shi et al., 2025). Naïve fine-tuning methods often cause *catastrophic forgetting*, where previously learned behaviors degrade in continual learning without access to original data (Kirkpatrick et al., 2017; Lopez-Paz & Ranzato, 2017; Luo et al., 2025). Recent empirical studies confirm that this phenomenon persists even in state-of-the-art models like Llama-3 and Qwen-2.5, often worsening as model scale increases (Jiang et al., 2024; Haque, 2025). Existing solutions, including regularization, replay-based, and retrieval-augmented methods, introduce computational overhead, require external memory, or are difficult to analyze theoretically (De Lange et al., 2021; Huang et al., 2024). Even small-scale updates can overwrite existing capabilities, raising concerns about stability and scalability (Meng et al., 2022; Zhai et al., 2023).

As modern LLMs become increasingly expressive, it is natural to question whether large-scale weight updates are necessary for acquiring a small amount of new knowledge. Therefore, in this work, we ask:

> *How can LLMs incorporate a small amount of new knowledge without catastrophic forgetting?*

To address this question, we follow a principled framework that models next-token generation in an LLM as a Markov process (Zekri et al., 2024; Yüksel & Flammarion, 2025), where tokens correspond to the states of the chain. Under this formulation, next-token probabilities coincide with the transition probabilities of the Markov chain, and the model's memory can be interpreted as a Markov transition matrix. Consequently, incorporating new knowledge without forgetting corresponds to expanding the state space of the Markov chain while preserving its original transition probabilities. To this end, we introduce a token-to-dictionary mapping strategy, under which new tokens are integrated by mapping them to a few existing tokens, without requiring adjustments to the original transition probabilities. Building on this Markov formulation and mapping strategy, we further

establish a sample complexity guarantee: the number of samples required to learn each new token scales as $O(s)$, where $s$ denotes the number of effective existing tokens to which the new token is mapped.

Embedding tuning, where only token embeddings are updated or expanded, provides a promising alternative for knowledge expansion (Liu et al., 2024; Wei et al., 2024). In this work, we establish connections between embedding tuning and the token-to-vocabulary mapping strategy. Intuitively, embedding updates preserve the model's transition structure because embeddings are token-specific representations that remain orthogonal to the token-transition computation graph, allowing new tokens to be mapped to the embedding dictionary without disrupting existing token-to-token dynamics.

This paper makes the following contributions:

- **Markovian formulation of language generation (§2).** We model language generation as a Markov process in which tokens correspond to states, and formalize the introduction of new tokens as an expansion of the Markov chain's state space. Under this perspective, eliminating catastrophic forgetting corresponds to preserving the transition dynamics among preexisting tokens. Additionally, we propose a mapping strategy under which new tokens are introduced while preserving the inherent transition behavior of the model.

- **Theoretical guarantees for vocabulary expansion (§3).** We establish sample complexity guarantees for learning new vocabulary, showing that semantic granularity governs data efficiency. Specifically, if a new token is mapped to $s$ preexisting tokens, the required number of samples scales as $O(s)$, which is independent of the original vocabulary size and model parameters. Our framework naturally extends to higher-order Markov chains (Section 3.3), where the sample complexity depends on the effective branching factor of natural language rather than the exponential growth of the vocabulary.

- **Empirical validation (§4).** To realize our mapping strategy in practice, we propose an embedding tuning algorithm in which only the embeddings corresponding to newly introduced tokens are updated. We evaluate this approach on controlled arithmetic-operator learning, synthetic vocabulary expansion, and real-world cross-lingual vocabulary expansion across Spanish, German, and Arabic. Across these settings, embedding tuning effectively integrates new tokens with fewer trainable parameters and induces little to no forgetting compared with full fine-tuning and other parameter-efficient baselines, further validating our theoretical findings.

The remainder of the paper is organized as follows. Section 2 presents the problem setup and preliminaries, and established connection between language models and Markov processes. Main theoretical results are presented in Section 3. Section 4 reports experimental results that validate our main claims and theoretical findings. Section 5 discusses related work. Finally, Section 6 concludes the paper and discusses limitations and future directions.

## 2. Problem Setup and Preliminaries

**Notation.** Let $[n]$ denote set $\{1, 2, \ldots, n\}$ for any positive integer $n$. Bold lowercase letters (e.g., $\mathbf{a}$) denote vectors with $\mathbf{a}_i$ being its $i$th coordinate. For any finite set $\mathcal{X}$, let $\Delta(\mathcal{X})$ denote the probability simplex over $\mathcal{X}$. Define sequence $x_{i:j}$ as $(x_{i+1}, x_{i+2}, \ldots, x_j)$. Let $D_{\mathrm{KL}}(\mathbf{p} \| \mathbf{q}) := \sum_i \mathbf{p}_i \log(\mathbf{p}_i / \mathbf{q}_i)$ be the Kullback–Leibler (KL) divergence. Additionally, we use $O(\cdot)$ to denote dependence up to constant factors, and $\tilde{O}(\cdot)$ suppress logarithmic factors.

### 2.1. Language Model as a Markov Process

Autoregressive language models generate text by sampling the next token from a probability distribution. This sequential generation mechanism induces a stochastic process and allows the model to be viewed as a Markov chain, as also discussed in (Zekri et al., 2024; Yüksel & Flammarion, 2025). Accordingly, in this work, we view next-token prediction as a Markovian state transition, with the model's conditional distributions defining the transition dynamics. Under this perspective, a pretrained language model can be represented as a Markov transition matrix over the token vocabulary.

Consider a pretrained large language model $\mathtt{LM}_{\theta}$. Let $\mathcal{V} := \{v_1, v_2, \cdots, v_T\}$ denote the token dictionary, which is a discrete finite set of size $T := |\mathcal{V}|$. For example, GPT-4 (Achiam et al., 2023) uses a BPE-based tokenizer (Gage, 1994) with a vocabulary size of approximately $T \approx 10^5$. Then given a prefix sequence $x_{t-k:t} := (x_{t-k+1}, \cdots, x_t) \in \mathcal{V}^k$ where $k$ denotes the active context window size, $\mathtt{LM}_{\theta}$ predicts the next token $x_{t+1} \in \mathcal{V}$ as

$$x_{t+1} \sim p_{\theta}\left( \cdot \mid x_{t-k:t} \right).$$

Here, $p_{\theta} \in \Delta(\mathcal{V})$ defines the model's output probability distribution conditioned on the context $x_{t-k:t}$, satisfying $\sum_{v \in \mathcal{V}} p_{\theta}(v \mid x_{t-k:t}) = 1$.

Note that, in this work, we assume that any decoding strategy (e.g., temperature scaling or top-$k$ truncation) has already been absorbed into $p_{\theta}$. Accordingly, we focus on standard sampling that draws tokens directly from $p_{\theta}$.

We follow a standard setting and approximate the language model with a first-order Markov chain defined on tokens.

Specifically, for any $v \in \mathcal{V}$, we have

$$p_{\boldsymbol{\theta}}(x_{t+1} = v \mid x_{t-k:t}) = p_{\boldsymbol{\theta}}(x_{t+1} = v \mid x_t). \quad (1)$$

Throughout the remainder of this paper, we denote by $\mathbf{p}^{(v)}$ the transition probability vector from a token $v$ to the vocabulary $\mathcal{V}$ of the pretrained language model with parameters $\boldsymbol{\theta}$. Specifically,

$$\mathbf{p}^{(v)} \in \Delta(\mathcal{V}) \quad \text{and} \quad \mathbf{p}_i^{(v)} = p_{\boldsymbol{\theta}}(x_{t+1} = v_i \mid x_t = v). \quad (2)$$

In practical language modeling, the conditional distribution of the next token typically depends on a sequence of preceding tokens, and the first-order assumption in (1) does not hold. More precisely, language models can be viewed as $K$-th order Markov chains, where $K$ denotes the maximum context length. We defer a detailed discussion to Section 3.3, where our first-order results can be directly applied by reconstructing the vocabulary set.

## 2.2. Knowledge Expansion

Knowledge expansion is essential for language models operating in dynamic real-world environments, where new words continually emerge after pretraining, including newly introduced or repurposed named entities (e.g., "COVID-19" as a virus, "DOGE" as a government department, and "X", formerly known as Twitter) as well as domain-specific terminology arising in scientific, medical, legal, and technical corpora. In this work, our focus is on *vocabulary expansion* as a concrete mechanism for achieving this goal: we introduce new tokens as lexical interfaces for new concepts, and study how these tokens can be integrated into the pretrained model while preserving its existing transition structure. Under the Markov chain formulation of language generation, incorporating new tokens corresponds to extending the state space of the Markov process.

Consider a set of new words/tokens $\mathcal{U} := \{u_1, u_2, \ldots, u_m\}$ where $|\mathcal{U}| = m \ll T$ and $\mathcal{U} \cap \mathcal{V} = \emptyset$. Introducing $\mathcal{U}$ to the model naturally expands the state space of the underlying Markov chain from $\mathcal{V}$ to $\mathcal{V} \cup \mathcal{U}$, and the model must learn how the new tokens interact with the existing ones in a coherent and semantically meaningful manner. Our goal is therefore to update the pretrained language model $\mathtt{LM}_{\boldsymbol{\theta}}$, with model parameters $\boldsymbol{\theta}$ updated to $\tilde{\boldsymbol{\theta}}$, such that the resulting transition distribution

$$p_{\tilde{\boldsymbol{\theta}}} : \mathcal{V} \cup \mathcal{U} \to \Delta(\mathcal{V} \cup \mathcal{U})$$

properly integrates the new tokens into the existing transition dynamics while preserving the original transitions among tokens in $\mathcal{V}$.

For any $u \in \mathcal{U}$, define its corresponding transition distribution over the existing vocabulary $\mathcal{V}$ under the updated parameters $\tilde{\boldsymbol{\theta}}$ as

$$\mathbf{q}^{(u)} \in \Delta(\mathcal{V}) \quad \text{and} \quad \mathbf{q}_i^{(u)} = p_{\tilde{\boldsymbol{\theta}}}(x_{t+1} = v_i \mid x_t = u). \quad (3)$$

Here, $\tilde{\boldsymbol{\theta}}$ corresponds to an oracle model (i.e., the optimal parameters achievable under infinite training data and a sufficiently expressive model) so that $p_{\tilde{\boldsymbol{\theta}}}$ represents the oracle transition distribution. We assume that 1) there are no transitions from existing tokens to new tokens so that the original transition structure is preserved; and 2) there are no transitions among new tokens, that is, $\sum_{v \in \mathcal{V}} p_{\tilde{\boldsymbol{\theta}}}(x_{t+1} = v \mid x_t = u) = 1$ for any $u \in \mathcal{U}$. Then the model's next-token distribution for $v_i \in \mathcal{V}$ satisfies

$$p_{\tilde{\boldsymbol{\theta}}}(x_{t+1} = v_i \mid x_t) = \begin{cases} \mathbf{p}_i^{(x_t)} & x_t \in \mathcal{V} \\ \mathbf{q}_i^{(x_t)} & x_t \in \mathcal{U} \end{cases},$$

where $\mathbf{p}^{(v)}, \mathbf{q}^{(u)} \in \Delta(\mathcal{V})$ are defined in (2) and (3), respectively. It guarantees that when the query token $x_t$ belongs to the original vocabulary $\mathcal{V}$, the predictive behavior of the updated model $\mathtt{LM}_{\tilde{\boldsymbol{\theta}}}$ remains identical to that of $\mathtt{LM}_{\boldsymbol{\theta}}$, thereby ensuring the absence of catastrophic forgetting. In contrast, when the query token is newly introduced ($x_t \in \mathcal{U}$), $\mathtt{LM}_{\tilde{\boldsymbol{\theta}}}$ admits it as a valid input and produces next-token predictions according to the prescribed oracle distribution $\mathbf{q}^{(u)} \in \Delta(\mathcal{V})$.

Note that in this work, we make the following assumptions:

**1. No transitions from existing tokens to new tokens** ($p_{\tilde{\boldsymbol{\theta}}}(u \mid v) = 0, \forall u \in \mathcal{U}, v \in \mathcal{V}$). We disregard transitions from tokens in $\mathcal{V}$ to tokens in $\mathcal{U}$. This reflects the assumption that, during language generation, if newly introduced concepts do not appear in the context, the model should not spontaneously generate new tokens. Instead, generation remains confined to the original vocabulary, thereby preserving the model's behavior on previously learned knowledge.

**2. No transitions among new tokens** ($p_{\tilde{\boldsymbol{\theta}}}(u_i \mid u_j) = 0, \forall u_i, u_j \in \mathcal{U}$). We also disregard transitions among tokens $u \in \mathcal{U}$. This assumption prevents newly introduced tokens from forming isolated subdynamics independent of the original vocabulary and ensures that they are integrated into the existing Markovian structure of the language model. If transitions among new tokens are desired, one may equivalently introduce new tokens sequentially, in which case the same analysis applies with the vocabulary size increasing as $T \to T + 1$ at each step.

**Extension beyond the assumptions.** The assumptions $p_{\tilde{\boldsymbol{\theta}}}(u \mid v) = 0$ and $p_{\tilde{\boldsymbol{\theta}}}(u_i \mid u_j) = 0$ are adopted to isolate forgetting and keep the sample-complexity analysis clean. More generally, allowing $\mathcal{V} \to \mathcal{U}$ introduces an additional dependence on the ingress probability into $\mathcal{U}$, while allowing $\mathcal{U} \to \mathcal{U}$ changes the complexity through the transition structure of the induced $\mathcal{U} \times \mathcal{U}$ subchain. We leave a full treatment of these more general cases to future work. Our experiments in Section 4 are less restrictive, since standard next-token training naturally allows both $\mathcal{V} \to \mathcal{U}$ and $\mathcal{U} \to \mathcal{U}$ transitions, which further extends the relevance of the theory to realistic settings.

## 2.3. Embeddings as Token Representations

Modern language models parameterize next-token transition distributions through a shared embedding space. In this view, token embeddings serve as continuous state representations of the underlying Markov process.

Let $\mathbf{E} \in \mathbb{R}^{T \times d}$ denote the embedding dictionary associated with the original vocabulary $\mathcal{V}$, where each token $v \in \mathcal{V}$ is mapped to an embedding vector $\mathbf{e}^{(v_i)} := \mathbf{E}_i \in \mathbb{R}^d$, and $d$ is the embedding dimension. The model's transition mechanism can then be expressed as a function $f : \mathbb{R}^d \to \Delta(\mathcal{V})$, which maps a token embedding to the corresponding next-token distribution. In particular, $f(\mathbf{e}^{(v)}) := \mathbf{p}^{(v)}$ for $v \in \mathcal{V}$.

**Mapping via embedding tuning.** Given a set of new tokens $\mathcal{U}$, we view embedding tuning as learning a token-to-dictionary mapping that represents each new token in terms of the original embedding dictionary $\mathbf{E}$.

**Definition 2.1.** For each $u \in \mathcal{U}$, let $\boldsymbol{\alpha}^{(u)} \in \mathbb{R}^T$ be the associated embedding representation/mapping such that $f(\mathbf{E}^\top \boldsymbol{\alpha}^{(u)}) = \mathbf{q}^{(u)}$.

Under this definition, learning a new token aims to identify a representation $\boldsymbol{\alpha}^{(u)}$ over the existing embedding dictionary $\mathbf{E}$ that induces the desired transition distribution. It formalizes how new tokens reuse and recombine semantic structure already encoded in the pretrained embeddings.

## 3. Main Results

In this section, we present our main theoretical results on embedding-based vocabulary expansion, where we analyze the sample complexity of learning new tokens under a token-to-dictionary mapping strategy. We begin by specifying the training dataset and its statistical assumptions.

**Definition 3.1** (Training Dataset). A training dataset consists of $N$ sequences $\mathcal{D} := \{(x_1^{(i)}, x_2^{(i)}, \cdots, x_{t_i}^{(i)})\}_{i=1}^N$ with tokens drawn from $\mathcal{V} \cup \mathcal{U}$. For each sequence $i \in [N]$, suppose that there exists at least one position $j < t_i$ such that $x_j^{(i)} \in \mathcal{U}$.

**Assumption 3.2.** Conditioned on $x_j^{(i)} \in \mathcal{U}$,

- the token is drawn uniformly from $\mathcal{U}$, i.e.,

$$\mathbb{P}(x_j^{(i)} = u \mid x_j^{(i)} \in \mathcal{U}) = 1/m \quad \text{for all } u \in \mathcal{U};$$

- the subsequent token is generated according to the oracle transition distribution, i.e.,

$$\mathbb{P}(x_{j+1}^{(i)} = v_{i'} \mid x_j^{(i)} = u) = \mathbf{q}_{i'}^{(u)} \quad \text{for all } u \in \mathcal{U}, \ i' \in [T].$$

**Assumption 3.3.** There exists a constant $c \in (0, 1]$ such that the output of $f$ is either zero or bounded below by $c$ in every coordinate, i.e., $f : \mathbb{R}^d \to (\{0\} \cup [c, 1])^T$.

Here, Assumption 3.2 ensures each sequence contains independent occurrences of new tokens, and the associated transitions follow the oracle model $p_{\tilde{\theta}}$. Assumption 3.3 guarantees the learnability of the output probability function $f$ with a finite number of samples. Otherwise, the sample complexity can scale as $O(\log^2 c)$ per token, which diverges as $c \to 0$. Note that similar lower-bounding conditions are implicitly used in practical LLM deployments through mechanisms such as probability truncation, top-$k$ sampling, and logit clipping, which effectively prevent extremely small probabilities from being realized.

### 3.1. One-to-one Token Mapping

We first consider the scenario where every new token $u \in \mathcal{U}$ behaves identically to one existing token in $\mathcal{V}$. Specifically, for any $u \in \mathcal{U}$, there exists $v^\star(u) \in \mathcal{V}$ such that

$$\mathbf{q}^{(u)} = \mathbf{p}^{(v^\star(u))}.$$

Under the mapping discussion from Definition 2.1, this is equivalent to $\boldsymbol{\alpha}^{(u)}$ being a one-hot vector.

**Definition 3.4** (KL Separation Margin). Define the separation margin with respect to the Kullback-Leibler divergence as

$$\gamma := \min_{i,j \in [T], i \neq j} D_{\mathrm{KL}}(\mathbf{p}^{(v_i)} \parallel \mathbf{p}^{(v_j)}).$$

The following theorem presents our main result for the one-to-one setting by establishing a sample complexity guarantee.

**Theorem 3.5.** *Suppose Assumptions 3.2 and 3.3 hold. Let $\gamma$ be the separation margin defined in Definition 3.4 and suppose $\gamma > 0$. To identify the correct token $v^\star(u)$ with probability at least $1 - \delta$, it suffices that*

$$N \geq \frac{8m}{\min\{\gamma^2 / \log^2 c, 1\}} \log \frac{2mT}{\delta}.$$

The proof is deferred to Appendix A.2, where we define the estimator $\hat{v}(u)$ via searching over $v \in \mathcal{V}$ to minimize the KL divergence:

$$\hat{v}(u) := \arg\min_{v \in \mathcal{V}} D_{\mathrm{KL}}(\hat{\mathbf{q}}^{(u)} \parallel \mathbf{p}^{(v)}).$$

Here, $\hat{\mathbf{q}}^{(u)}$ denotes the empirical distribution, with formal definition deferred to (4).

Theorem 3.5 implies that the sample complexity per new token scales as $\tilde{O}(\gamma^{-2} \log^2 c)$ (up to logarithmic factors). In particular, the dominant rate depends only on the KL separation margin $\gamma$ and the lower-bound parameter $c$. This highlights that, in the one-to-one setting, identifying $v^\star(u)$ is governed by distributional separation rather than the original vocabulary size $T$ and the embedding dimension $d$.

A natural concern is that $\gamma$ may vanish as the vocabulary size $T$ grows. However, this behavior is not inherent. In practice, token transition distributions often exhibit clustering or grouping structure, in which case the minimal KL separation between different clusters remains bounded below by a constant. Concretely, when tokens are organized into groups of size $O(b)$, the effective separation margin $\gamma$ then scales independent of $T$. Moreover, common sampling procedures such as top-$K$ further reinforce this by concentrating probability mass on a small subset of likely tokens.

### 3.2. New Token as a Sparse Combination

The one-to-one setting in Section 3.1 considers an extreme case where each new token exactly recovers a single oracle distribution $\mathbf{p}^{(v^\star(u))}$. In practice, however, a new token may behave, in the transition dynamics, like a structured combination of several similar existing tokens. In this section, we relax this assumption and consider more general scenarios in which a new token corresponds to a combination of multiple existing tokens.

**Assumption 3.6.** For a given sparsity level $s$ and a constant $B < \infty$, we assume that each new token $u \in \mathcal{U}$ admits a representation vector $\boldsymbol{\alpha}^{(u)}$ (defined in Definition 2.1) satisfying $\boldsymbol{\alpha}^{(u)} \in \mathcal{A}$ where $\mathcal{A} := \{\boldsymbol{\alpha} \in \mathbb{R}^T : \|\boldsymbol{\alpha}\|_0 \leq s, \|\boldsymbol{\alpha}\|_2 \leq B\}$.

Assumption 3.6 restricts each representation vector $\boldsymbol{\alpha}^{(u)}$ to be $s$-sparse, reflecting that a new token can be expressed as a mixture of at most $s$ existing tokens in $\mathcal{V}$. While the case $s \approx T$ is generally unrealistic, our result in Theorem 3.7 still holds (by replacing $s$ with $T$).

Consider the training dataset $\mathcal{D}$ as in Definition 3.1. For each $u \in \mathcal{U}$, we define the empirical transition distribution $\hat{\mathbf{q}}^{(u)} \in \Delta(\mathcal{V})$ by

$$\hat{\mathbf{q}}_k^{(u)} = \frac{\sum_{i \in [N]} \sum_{j \in [t_i]} \mathbf{1}\{x_j^{(i)} = u, x_{j+1}^{(i)} = v_k\}}{\sum_{i \in [N]} \sum_{j \in [t_i]} \mathbf{1}\{x_j^{(i)} = u\}} \quad \text{for } k \in [T].$$

(4)

Following the embedding representation in Definition 2.1, we estimate the representation vectors $\{\hat{\boldsymbol{\alpha}}^{(u)}\}_{u \in \mathcal{U}}$ by solving

$$\hat{\boldsymbol{\alpha}}^{(u)} := \arg \min_{\boldsymbol{\alpha}^{(u)} \in \mathcal{A}} D_{\mathrm{KL}}(\hat{\mathbf{q}}^{(u)} \| f(\mathbf{E}^\top \boldsymbol{\alpha}^{(u)})) \quad \text{for } u \in \mathcal{U}. \quad (5)$$

The estimation risk is evaluated via the worst-case KL divergence between the ground-truth distribution $\mathbf{q}^{(u)}$ (see (3) and Definition 2.1) and the estimated distribution induced by $\hat{\boldsymbol{\alpha}}^{(u)}$ from (5).

$$\mathcal{R}(\{\hat{\boldsymbol{\alpha}}^{(u)}\}_{u \in \mathcal{U}}) := \max_{u \in \mathcal{U}} D_{\mathrm{KL}}(\mathbf{q}^{(u)} \| f(\mathbf{E}^\top \hat{\boldsymbol{\alpha}}^{(u)})). \quad (6)$$

We now state our main result for the sparse mixture setting.

**Theorem 3.7.** *Suppose Assumptions 3.2, 3.3 and 3.6 hold, and suppose $f : \mathbb{R}^d \rightarrow \Delta(\mathcal{V})$ is $L$-Lipschitz satisfying $\|f(\mathbf{x}) - f(\mathbf{y})\|_1 \leq L\|\mathbf{x} - \mathbf{y}\|_2$. If the sample size satisfies*

$$N \geq O\left(m \cdot \log \frac{m}{\delta}\right),$$

*then with probability at least $1-\delta$, the estimation risk defined in (6) satisfies*

$$\mathcal{R}(\{\hat{\boldsymbol{\alpha}}^{(u)}\}_{u \in \mathcal{U}}) \leq \min_{\varepsilon > 0} \left\{\varepsilon + O\left(\sqrt{\frac{m \log^2 c}{N}\left(\log \frac{m}{\delta} + s \log \frac{T}{cs\varepsilon}\right)}\right)\right\}.$$

*Here, $O(\cdot)$ subsumes constant factors depending on $L$, $B$, and the maximal singular value of the embedding matrix $\mathbf{E}$.*

We defer the detailed bound and its proof to Appendix A.3. Theorem 3.7 shows that learning the new knowledge $\mathcal{U}$ of size $m = |\mathcal{U}|$ does not require a number of samples proportional to the full vocabulary size $T = |\mathcal{V}|$ or the model dimensionality. Instead, it depends only on the sparsity of the token-to-dictionary representations: for each new token, it suffices to have $\tilde{O}(s \log^2 c)$ samples.

### 3.3. Higher-order Markov Chain

In this section, we consider the practical scenario where next-token prediction depends on a sequence of prefix tokens (rather than a single token), and model the prediction dynamics as a higher-order Markov process.

Recall the discussion in Section 2.1. Consider a language model with maximal window size $K$. Then next-token distribution $p_\theta$ induced by $\mathrm{LM}_\theta$ is equivalent to a $K$-order Markov chain. Specifically, let $x_{t-K+1}, \cdots, x_t, x_{t+1} \in \mathcal{V} \cup \{\mathtt{NULL}\}$ where "$\mathtt{NULL}$" denotes a special padding token used when the available context is shorter than $K$. Define the augmented state space $\mathcal{V}' = (\mathcal{V} \cup \{\mathtt{NULL}\})^K$. Then it results in a first-order Markov chain over $\mathcal{V}'$ with the corresponding transition matrix given by

$$\mathbf{p}^{(x_{t-K:t})} = p_\theta(\cdot \mid x_{t-K:t}) \in \Delta(\mathcal{V}').$$

Under this construction, all first-order results apply directly by replacing the original vocabulary $\mathcal{V}$ with $\mathcal{V}'$. Accordingly, the new token set is extended from $\mathcal{U}$ to $\mathcal{U}' := (\mathcal{V} \cup \{\mathtt{NULL}\})^{K-1} \times \mathcal{U}$, which consists of length-$K$ contexts whose final token is newly introduced.

At first glance, $|\mathcal{V}'| \approx T^K$ appears prohibitively large. However, in practice, only a small subset of these sequences is syntactically and semantically meaningful. For example:

> ✓ *The cat sat on the mat.*
> ✗ *Mat the on sat cat the.*

Here, the first sentence represents a valid linguistic context, whereas the second does not. This motivates restricting

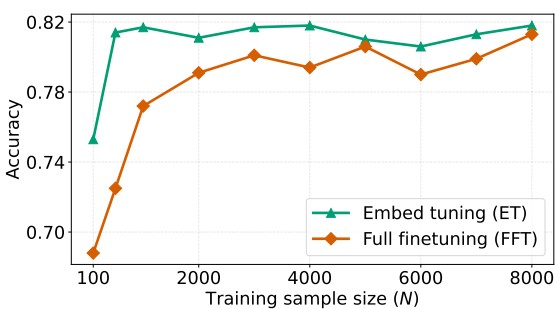

*Figure 1.* Accuracy of the special token operation $a\langle\text{spec}\rangle b = a \times b$ across different training sample sizes $N$. The ET method consistently outperforms FFT in low-data regimes, demonstrating its improved sample efficiency.

*Table 1.* Accuracy on $a\langle\text{spec}\rangle b$, $a * b$, and $a + b$ under embedding tuning (ET) and full fine-tuning (FFT) when $N = 1000$. While FFT improves performance on multiplication, it causes catastrophic forgetting on addition, collapsing accuracy from 100% to 0%. In contrast, ET preserves performance on all pre-existing tasks, demonstrating zero forgetting.

| Method | Accuracy | Forgetting | | |
|---|---|---|---|---|
| | $a\langle\text{spec}\rangle b=ab$ | $a*b=ab$ | $a+b=a+b$ | $a+b=ab$ |
| Base model | 4.8% | 63.5% | 100.0% | 0.0% |
| FFT | 77.2% | 77.0% | 0.0% | 76.2% |
| ET | 81.4% | 63.7% | 100.0% | 0.0% |

attention to a much smaller set of "meaningful" states $\bar{\mathcal{V}} \subset \mathcal{V}'$, with $|\bar{\mathcal{V}}| \ll |\mathcal{V}'|$.

A reasonable estimate is $|\bar{\mathcal{V}}| = O(Tb^{K-1})$ where $b \ll T$ represents the effective branching factor of natural language, constrained by syntax, semantics, etc. For instance, the number of valid words that can follow "*cat*" is only $O(b)$, rather than the full vocabulary size $T$. Accordingly, this leads to an effective size $O(mb^{K-1})$ for the new token contexts. Following our discussion in Section 3.2, this suggests a sparsity level of approximately $s = O(Kb)$.

Additionally, although the transition matrix is of size $O(Tb^{K-1}) \times O(Tb^{K-1})$, the same linguistic constraints imply that each transition probability vector $(\mathbf{p}^{(\cdot)})$ is highly sparse, containing only $O(b)$ nonzero entries.

## 4. Experiments

### 4.1. New Tokens as Arithmetic Operators

To empirically validate the Markovian framework for knowledge expansion, we design a controlled experiment in which a semantically-null token (denoted by $\langle\text{spec}\rangle$) is mapped to a precise functional role (arithmetic multiplication), e.g., $a\langle\text{spec}\rangle b = a \times b$. The extended vocabulary becomes $\mathcal{U} = \{\langle\text{spec}\rangle\}$ with $|\mathcal{U}| = m = 1$. This setup enables a clean evaluation of two central properties of our token-to-dictionary mapping strategy: (i) *sample efficiency* in recovering the correct operator, and (ii) strict *preservation* of pre-existing computational structure.

**Experimental setup.** We conduct all the experiments in this section using the Llama-3.2-3B-Instruct model (Meta AI, 2024b). Let $\langle\text{spec}\rangle$ denote a previously unseen special token, implemented as `<|reserved_special_token_0|>` in Llama-3.2. We then construct an arithmetic dataset in which each (input, output) pair takes the form of ("$a\langle\text{spec}\rangle b$ =", "$a \times b$"), where $a, b \leq 100$ are positive integers and $\langle\text{spec}\rangle$ is intended to function as a multipli-

cation operator. For example, the input "$20\langle\text{spec}\rangle 26$ =" corresponds to the target output "520". In our experiments, we vary the number of training samples $N$ from 100 to 8000. All results are evaluated on a fixed test set of 1000 examples, which is strictly disjoint from the training data.

To realize the token-to-vocabulary mapping strategy as discussed in Section 2.3 and following Definition 2.1, we propose a simple *embedding tuning* (ET) optimization strategy, where during training, only the embedding vector corresponding to the special token $\langle\text{spec}\rangle$ is trained, resulting in only 3,072 trainable parameters out of approximately 3B total parameters. For comparison, we consider the standard *full fine-tuning* (FFT) where all model parameters are updated. The experimental results are presented in Figure 1.

• **Evidence of sample efficiency.** Figure 1 illustrates the prediction accuracy of the new token $\langle\text{spec}\rangle$ when used as a multiplication operator across a wide range of training sample sizes $N$. We observe that embedding tuning consistently outperforms full fine-tuning across all regimes. Notably, embedding tuning achieves strong performance with as few as $N \approx 500$ training samples. Based on our Markovian framework as described in Sections 2 and 3, embedding tuning directly optimizes the transition vector associated with the new token by mapping it into the existing tokens that are semantically related to the multiplication operation (e.g., the token '$*$').

• **Evidence of zero forgetting.** To evaluate catastrophic forgetting after fine-tuning, we track the performance changes on both multiplication ($a * b$) and addition ($a + b$). The results are summarized in Table 1, from which we make the following observations:

1. FFT increases the base model's accuracy on the standard multiplication task $a * b$, whereas ET preserves the original accuracy (1st column of Forgetting). This behavior is expected: FFT updates all model parameters, so learning $\langle\text{spec}\rangle$ also alters existing arithmetic representations. In contrast, ET leaves the original

transitions unchanged and therefore preserves the base model's performance.

2. We further evaluate performance on the addition task $a+b$, for which the base model achieves 100% accuracy on the test set. Under FFT, the addition accuracy collapses from 100% to 0%, indicating severe catastrophic forgetting: while learning ⟨spec⟩ as a multiplication operator, the model entirely loses its ability to perform addition. In contrast, ET maintains 100% accuracy, demonstrating zero forgetting.

3. To better understand the behavior of FFT, we additionally measure the accuracy when the model is evaluated under the criterion that the output of $a + b$ equals $a \times b$ rather than $a + b$. As shown in the last column of Table 1, this accuracy closely matches that of both $a$⟨spec⟩$b$ and $a * b$. This indicates that FFT has overwritten the original arithmetic manifold, causing the model to systematically apply multiplicative logic even to addition prompts.

• **Implicit token-to-dictionary mapping.** The results in Table 1 show that after applying embedding tuning, the accuracy of the new token operation $a$ ⟨spec⟩ $b$ (81.4%) exceeds that of the standard multiplication token $a * b$ (63.5%). To investigate the underlying reason, we consider alternative contexts that also express multiplication. Specifically, the base model achieves individual accuracies of 63.5% for "$a * b$", 65.2% for "$a \times b$", 60.6% for "$a$ times $b$", and 58.4% for "$a$ multiplies $b$". While each formulation yields only moderate accuracy, their ensemble accuracy reaches 73.2%. This explains why the final $a$ ⟨spec⟩ $b$ accuracy surpasses that of the token $*$. Moreover, it provides empirical support for the sparse combination hypothesis introduced in Section 3.2: rather than learning a one-to-one mapping to the '$*$' token, the embedding of ⟨spec⟩ is optimized as a sparse representation over a semantic dictionary of multiplication-related states, leading to improved performance.

### 4.2. Vocabulary Expansion with Synthetic Words

In this section, we run experiments to evaluate the ability of language models to efficiently integrate *novel synthetic vocabulary* without forgetting previously learned knowledge.

**Experimental setup.** We construct a benchmark consisting of *100 synthetic words/tokens*, corresponding to $\mathcal{U} = \{u_1, u_2, \cdots, u_{100}\}$ where $|\mathcal{U}| = 100$. These words are injected into language sentences following natural language sentences using Phi-3.5 Mini Instruct (Microsoft, 2024), ensuring that the surrounding linguistic context remains realistic while the token itself carries no prior semantic meaning. As a running example, consider the following pair of sentences:

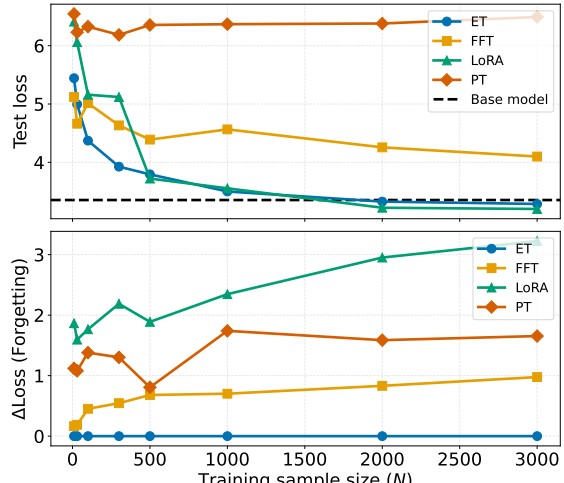

*Figure 2.* Synthetic vocabulary learning and forgetting. **Top:** Test loss versus the number of synthetic training sentences $N$; the dashed line denotes the base model's performance on real sentences. **Bottom:** Forgetting across methods and sample sizes, where embedding tuning achieves zero forgetting while other methods incur increasing forgetting as $N$ grows. Code repo: Synthetic-vocab-expansion

**Real:** *The fiber in broccoli helps regulate blood sugar levels.*

**Synthetic:** *The fiber in glor helps regulate blood zorp levels.*

Here, we treat "broccoli" and "sugar" as real tokens/words, whereas "glor" and "zorp" are newly introduced fake tokens with no prior occurrences in the pretraining corpus. More details on synthetic sentences generation are discussed in Appendix B.1. To assess catastrophic forgetting, we concurrently evaluate model performance on the WikiText database. We define forgetting as the drop in WikiText performance, measured by the increase in loss before and after adapting the model to the synthetic sentences.

In addition to the embedding tuning (ET) and full fine-tuning (FFT) algorithms described in Section 4.1, we also consider *low-rank adaptation* (LoRA) and *prompt tuning* (PT) in this section. For prompt tuning, a fixed number of *continuous* prompt embeddings are prepended to the input sequence and optimized, while the backbone model remains frozen.

In Figure 2, we vary the number of synthetic training sentences ($N$) from 10 to 3000 and tune the LLaMA-3.2-1B model (Meta AI, 2024a) for 100 epochs. Performance is evaluated using token-level cross-entropy loss on a fixed test set of 1000 synthetic sentences. Specifically, Figure 2 (top) reports the test loss, where the horizontal black dashed line indicates the base model's performance on real sentences. As the number of training samples increases (e.g., $N = 3000$), the test loss converges to this baseline. Figure 2 (bottom) shows the extent of forgetting under different training algorithms and sample sizes. Consistent with our

theoretical findings, embedding tuning, which corresponds to the token-to-dictionary mapping strategy, achieves strong performance without inducing any forgetting. In contrast, forgetting is unavoidable for the otherthree methods and becomes more pronounced as the number of training samples increases.

Additional results are reported in Table 2, where we evaluate seven pretrained models using $N = 1000$ synthetic training sentences. Base Wiki loss denotes the token-level cross-entropy on the WikiText dataset for each pretrained model, and forgetting is measured as the increase in WikiText loss after tuning. Among all methods, ET achieves a competitive test loss while inducing zero forgetting on WikiText, confirming that restricting updates to newly introduced token embeddings enables effective acquisition of novel vocabulary without disrupting previously learned language representations. In contrast, FFT and LoRA exhibit substantially higher test loss and severe forgetting, with loss increases exceeding 8.0 in some settings. PT attains competitive test loss for certain models, such as Phi-3.5 Mini Instruct (Microsoft, 2024) but shows inconsistent forgetting behavior across architectures. A likely explanation is that the synthetic dataset is generated using Phi-3.5 Mini Instruct (Microsoft, 2024) itself, so prompt tuning can be particularly well aligned with the data generation process and thus yield lower loss. Overall, these results demonstrate that ET provides the most favorable trade-off between sample-efficient vocabulary expansion and strict preservation of prior knowledge, empirically validating our theoretical findings. To further assess practical relevance beyond synthetic vocabulary injection, we next evaluate the same adaptation methods in real-world cross-lingual vocabulary-expansion settings ranging from relatively close language pairs (Spanish and German) to a substantially more distant one (Arabic).

### 4.3. Cross-lingual Vocabulary Expansion

To complement the controlled synthetic setting in Section 4.2, we additionally evaluate our method in a real-world cross-lingual vocabulary-expansion setting. We consider three target languages with different degrees of distance from English: Spanish and German as relatively closer language pairs, and Arabic as a substantially more distant one. This progression allows us to assess whether the proposed method continues to provide strong adaptation and low forgetting as the lexical gap increases.

**Experimental setup.** For each target language, namely Spanish, German, and Arabic, we construct monolingual corpora from Wikipedia and retain examples with at least 200 characters to ensure sufficient local context. We use the same data split for all three target languages: 6,000 target-language training examples, 1,000 held-out target-language test examples, and 1,000 held-out English

Wikipedia examples for measuring source-language retention. In the reported experiments, we use $N_{\text{train}} = 1000$ target-language training examples for adaptation, and evaluate target-language acquisition and English forgetting on the corresponding held-out test sets. All examples are tokenized with a maximum sequence length of 256 tokens. We use Qwen2.5-3B as the backbone model (Qwen Team, 2024).

The results are reported in Table 3. Across Spanish, German, and Arabic, ET achieves the strongest overall trade-off between target-language adaptation and English retention. In particular, ET obtains the lowest target-language loss in the Qwen2.5-3B (Qwen Team, 2024) experiments while maintaining near-zero, and in some cases slightly negative, English forgetting. These results show that the proposed token-to-dictionary mapping strategy extends beyond synthetic settings and remains effective in realistic vocabulary-expansion scenarios. The slightly negative forgetting values are an interesting observation. This may be because Spanish and German are more closely aligned with English than Arabic, and learning these languages could therefore enhance the model's original performance. The effect may also depend on the backbone model, training dynamics, tokenizer, and evaluation set. Further exploration of this phenomenon would be valuable. Additional Arabic results across Qwen2.5 (Qwen Team, 2024) and Llama3.2 (Meta AI, 2024b) backbones are also provided in Appendix B.2.

## 5. Related Work

**Continual Learning and Forgetting.** Catastrophic forgetting remains a central challenge in adapting large language models. Classical continual learning methods, such as EWC, GEM, and A-GEM, mitigate forgetting in smaller models but are difficult to scale effectively to modern LLMs (Kirkpatrick et al., 2017; Lopez-Paz & Ranzato, 2017; Chaudhry et al., 2019; Wu et al., 2024; De Lange et al., 2021; Shi et al., 2025). Parameter-efficient fine-tuning methods, including LoRA, prefix tuning, and adapters, reduce the number of updated parameters, but do not by themselves guarantee retention of prior behavior (Hu et al., 2021; Li & Liang, 2021; Liu et al., 2022; Pfeiffer et al., 2021; Gupta et al., 2024; Bafghi et al., 2024). Recent methods such as OPLoRA and MoFO further reduce interference through constrained or sparse updates (Xiong & Xie, 2025; Chen et al., 2024), but a general theoretical account of when sparse updates prevent forgetting remains limited.

**LLMs as Markov Processes.** Autoregressive LLMs can be viewed as Markov processes in which next-token probabilities define transition dynamics over tokens or token histories (Zekri et al., 2025; Ildiz et al., 2024). Related work has also studied transformers on Markov data, MCMC-based latent reasoning, and Markov-process interpretations of reasoning and RLHF (Rajaraman et al., 2024; Hoffman et al., 2023;

*Table 2.* Comparison of adaptation methods across models with $N = 1000$ synthetic training sentences, presenting test loss and forgetting (measured as WikiText loss increase after tuning) across different pretrained models and tuning methods.

| Model | Performance (loss) | | | | Forgetting | | | | |
|---|---|---|---|---|---|---|---|---|---|
| | FFT | LoRA | PT | ET | Base Wiki loss | FFT | LoRA | PT | ET |
| Phi-3.5 Mini Instruct | 3.28 | 6.01 | **1.87** | 2.75 | 2.43 | +9.10 | +8.80 | +0.39 | **.00** |
| Llama3.2-1B | 4.57 | 3.55 | 6.37 | **3.50** | 2.86 | +0.70 | +2.34 | +1.56 | **.00** |
| Llama3.2-3B | 4.54 | 4.08 | 5.76 | **3.47** | 2.62 | +0.51 | +1.35 | +0.62 | **.00** |
| Llama3.1-8B | 4.40 | 7.63 | 5.86 | **3.79** | 2.42 | +1.24 | +8.36 | +0.69 | **.00** |
| Qwen2.5-1.5B | 4.81 | 3.72 | 4.03 | **3.46** | 2.73 | +11.59 | +1.60 | +2.19 | **.00** |
| Qwen2.5-3B | 5.14 | 3.45 | 4.71 | **3.39** | 2.59 | +4.71 | +1.13 | +0.06 | **.00** |
| Qwen2.5-7B | 5.22 | 3.86 | 4.62 | **3.37** | 2.43 | +7.37 | +8.60 | +0.01 | **.00** |

*Table 3.* English-to-Spanish, English-to-German, and English-to-Arabic vocabulary expansion results on Qwen2.5-3B. Forgetting is measured as the increase in English loss after adaptation.

| Target language | Performance (loss) | | | | Forgetting | | | | |
|---|---|---|---|---|---|---|---|---|---|
| | FFT | LoRA | PT | ET | Base Eng. loss | FFT | LoRA | PT | ET |
| Spanish | 5.56 | 3.17 | 3.45 | **2.30** | 2.01 | +9.83 | +0.66 | -0.03 | **-0.04** |
| German | 6.80 | 3.29 | 3.63 | **3.27** | 2.01 | +9.81 | +0.49 | -0.03 | **-0.08** |
| Arabic | 7.95 | 3.95 | 4.34 | **2.82** | 2.01 | +11.95 | +0.46 | +0.06 | **0.00** |

Zhang et al., 2025; Kim et al., 2025). These perspectives motivate our use of a Markov transition framework for analyzing stability, forgetting, and knowledge integration in vocabulary expansion.

**Vocabulary Expansion.** Our work is related to vocabulary expansion, cross-lingual adaptation, and sparse representation learning. Prior work shows that pretrained models can be extended to new languages or lexical inventories by learning target-language embeddings, shallow alignment layers, or language-specific modules (Artetxe et al., 2020; Marchisio et al., 2023; Pfeiffer et al., 2020; 2022). Domain and multimodal adaptation methods have also studied explicit vocabulary expansion, including selective domain-token addition and vision-vocabulary scaling (Liu et al., 2024; Wei et al., 2024; Yamaguchi et al., 2025b). In contrast, we formulate vocabulary expansion in autoregressive LMs as state-space expansion in a Markov process and analyze the sample complexity of learning new tokens through token-to-dictionary mappings. This connects vocabulary expansion to sparse coding, where the cost of learning a new token depends on the sparsity of its representation rather than the full vocabulary size (Olshausen & Field, 1997).

**Knowledge Retention via Embedding Updates.** Several lines of work aim to preserve prior knowledge during model adaptation, including model editing, sparse updates, and source-shielded updates (Meng et al., 2022; Gupta et al., 2024; Chen et al., 2025; Yamaguchi et al., 2025a). Another related direction improves language plasticity during pretraining, for example through active forgetting by periodically resetting embeddings (Chen et al., 2023). Our

setting differs from these approaches: we study post-training vocabulary expansion, where only newly introduced token representations are updated. Under our Markovian formulation, this preserves the original vocabulary-to-vocabulary transition structure, yielding a formal zero-forgetting guarantee for the original token space.

## 6. Discussion

Our work links language generation to Markov transition dynamics, allowing new knowledge expansion to be formalized as an expansion of the state space of an existing Markov chain. We consider the setting in which newly introduced tokens can be mapped to a small number of pre-existing tokens. Under this setting, (i) learning new tokens is sample-efficient, with the required number of samples scaling linearly with the number of mapped tokens, and (ii) no forgetting is introduced through this integration. We further propose an embedding tuning algorithm that implements this mapping strategy in practice. Both theoretical guarantees and empirical evidence support our claims.

**Limitations and future directions.** Our theoretical analysis assumes that newly introduced tokens occur with equal probability. In practice, token frequencies are more complex and typically depend on the existing tokens to which they are mapped. Moreover, our framework assumes sufficiently expressive LLMs. An interesting direction for future work is to study the scaling behavior when model capacity is limited, as well as to explore trade-offs among different adaptation algorithms under finite-capacity regimes.

## Impact Statement

This work studies methods for incorporating new vocabulary and domain-specific knowledge into large language models while reducing catastrophic forgetting. The potential positive impacts include more efficient model adaptation, lower computational cost for vocabulary expansion, and improved stability when updating models for new languages, domains, or emerging terminology. Potential risks include misuse for rapidly adapting models to misleading, harmful, or privacy-sensitive terminology if training data is not properly curated. Our experiments use synthetic and public text corpora, and deployment of such methods should include auditing for data quality, privacy, bias, and downstream misuse.

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

# A. Proofs in Section 3

## A.1. Supporting Lemma

**Lemma A.1.** *Consider the training dataset as in Definition 3.1. Suppose Assumption 3.2 holds. Define the minimum empirical count*

$$N_{\min} := \min_{u \in \mathcal{U}} N_u \quad where \quad N_u = \sum_{i \in [N], j \in [t_i]} \mathbf{1}\{x_j^{(i)} = u\}. \tag{7}$$

*Recall $m = |\mathcal{U}|$. Then for any $\varepsilon \in (0, 1)$, we have for any $u \in \mathcal{U}$,*

$$\mathbb{P}\left(N_u \geq (1 - \varepsilon)\frac{N}{m}\right) \geq 1 - \exp\left(-\frac{\varepsilon^2}{2} \cdot \frac{N}{m}\right). \tag{8}$$

*and*

$$\mathbb{P}\left(N_{\min} \geq (1 - \varepsilon)\frac{N}{m}\right) \geq 1 - m \cdot \exp\left(-\frac{\varepsilon^2}{2} \cdot \frac{N}{m}\right). \tag{9}$$

*Proof.* Recall the count of token $u$ in the dataset from (7), denoted by $N_u$. We form a dataset consisting of all tokens in the training dataset $\mathcal{D}$, satisfying $x_j^{(i)} \in \mathcal{U}$, and denote it by $\{z_i\}_{i=1}^{N'}$ where $z_i \in \mathcal{U}$ and $N' = \sum_{u \in \mathcal{U}} N_u \geq N$.

Fix any $u \in \mathcal{U}$. Under Assumption 3.2, the random variables $\mathbf{1}\{z_i = u\}$ are i.i.d. Bernoulli with $\mathbb{P}(z_i = u) = 1/m$. Hence,

$$N_u \sim \text{Binomial}\left(N', \frac{1}{m}\right), \qquad \mathbb{E}[N_u] = \frac{N'}{m}.$$

By the Chernoff bound and following (Tsun, 2020), for any $\varepsilon \in (0, 1)$,

$$\mathbb{P}\left(N_u \leq (1 - \varepsilon)\frac{N'}{m}\right) \leq \exp\left(-\frac{\varepsilon^2}{2}\frac{N'}{m}\right).$$

Since $N \leq N'$, we get

$$\mathbb{P}\left(N_u \leq (1 - \varepsilon)\frac{N}{m}\right) \leq \mathbb{P}\left(N_u \leq (1 - \varepsilon)\frac{N'}{m}\right) \leq \exp\left(-\frac{\varepsilon^2}{2}\frac{N'}{m}\right) \leq \exp\left(-\frac{\varepsilon^2}{2}\frac{N}{m}\right).$$

It proves (8). Now applying the union bound over all $u \in \mathcal{U}$, we get

$$\mathbb{P}\left(N_{\min} \leq (1 - \varepsilon)\frac{N}{m}\right) = \mathbb{P}\left(\exists u \in \mathcal{U} : N_u \leq (1 - \varepsilon)\frac{N}{m}\right) \leq \sum_{u \in \mathcal{U}} \mathbb{P}\left(N_u \leq (1 - \varepsilon)\frac{N}{m}\right) \leq m \cdot \exp\left(-\frac{\varepsilon^2}{2}\frac{N}{m}\right).$$

It proves (9). $\qquad\square$

## A.2. Proof of Theorem 3.5

*Proof.* Recall from Definition 3.1, where we are given a training dataset

$$\mathcal{D} := \{(x_1^{(i)}, x_2^{(i)}, \cdots, x_{t_i}^{(i)})\}_{i=1}^N.$$

**Step 1: Define the optimization objective and the algorithm for finding the optimal $\hat{v}(u)$ (for each $u \in \mathcal{U}$).**

For any $u \in \mathcal{U}$, define its number of occurrences as

$$N_u = \sum_{i \in [N]} \sum_{j \in [t_i]} \mathbf{1}\{x_j^{(i)} = u\},$$

where we have $\sum_{u \in \mathcal{U}} N_u \geq N$ following Definition 3.1.

Define its corresponding empirical distribution $\hat{\mathbf{q}}^{(u)} \in \Delta(\mathcal{V})$ (similar to (4)) as

$$\hat{\mathbf{q}}_k^{(u)} = \frac{\sum_{i \in [N]} \sum_{j \in [t_i]} \mathbf{1}\{x_j^{(i)} = u, x_{j+1}^{(i)} = v_k\}}{\sum_{i \in [N]} \sum_{j \in [t_i]} \mathbf{1}\{x_j^{(i)} = u\}} \quad \text{for } k \in [T]. \tag{10}$$

For simplicity and without loss of generality, let $\hat{\mathbf{q}} = \hat{\mathbf{q}}^{(u)}$, $\mathbf{q}^\star = \mathbf{q}^{(u)}$, $\mathbf{p}^\star = \mathbf{p}^{(v^\star(u))}$, $\hat{v} = \hat{v}(u)$ and $v^\star = v^\star(u)$. Here considering the one-to-one mapping in Section 3.1 where $\mathbf{q}^{(u)} = \mathbf{p}^{(v^\star(u))}$, we also have $\mathbf{q}^\star = \mathbf{p}^\star$. Additionally, rewrite the collected data corresponding to $u$ as $\{z_i\}_{i=1}^{N_u}$ where following Assumption 3.2, we have that $z_i \in \mathcal{V}$ is i.i.d. sampled from $\mathbf{q}^\star$. Additionally, let $\{\text{ID}_i\}_{i=1}^{N_u}$ ($\text{ID}_i \in [T]$) be the corresponding token ID's of $\{z_i\}_{i=1}^{N_u}$ in $\mathcal{V}$ where $z_i = v_{\text{ID}_i}$ for $i \in [N_u]$.

Consider the estimator for token $u$

$$\hat{v} = \arg \min_{v \in \mathcal{V}} D_{\text{KL}}(\hat{\mathbf{q}} \| \mathbf{p}^{(v)})$$

where we have

$$\arg \min_{v \in \mathcal{V}} D_{\text{KL}}(\hat{\mathbf{q}} \| \mathbf{p}^{(v)}) = \arg \min_{v \in \mathcal{V}} \sum_{i \in [T], \hat{\mathbf{q}}_i \neq 0} \hat{\mathbf{q}}_i \log \frac{\hat{\mathbf{q}}_i}{\mathbf{p}_i^{(v)}}$$

$$= \arg \min_{v \in \mathcal{V}} \left( \underbrace{\sum_{i \in [T], \hat{\mathbf{q}}_i \neq 0} \hat{\mathbf{q}}_i \log \hat{\mathbf{q}}_i}_{\text{constant}} - \sum_{i \in [T], \hat{\mathbf{q}}_i \neq 0} \hat{\mathbf{q}}_i \log \mathbf{p}_i^{(v)} \right)$$

$$= \arg \min_{v \in \mathcal{V}} \sum_{i \in [T]} -\hat{\mathbf{q}}_i \log \mathbf{p}_i^{(v)}$$

$$= \arg \min_{v \in \mathcal{V}} \sum_{i \in [T]} -\frac{\sum_{j \in [N_u]} \mathbf{1}\{z_j = v_i\}}{N_u} \log \mathbf{p}_i^{(v)}$$

$$= \arg \min_{v \in \mathcal{V}} \sum_{j \in [N_u]} \sum_{i \in [T]} -\mathbf{1}\{z_j = v_i\} \log \mathbf{p}_i^{(v)}$$

$$= \arg \min_{v \in \mathcal{V}} \sum_{i \in [N_u]} -\log \mathbf{p}_{\text{ID}_i}^{(v)}.$$

Then we can rewrite the estimator via

$$\hat{v} = \arg \min_{v \in \mathcal{V}} \sum_{i \in [N_u]} -\log \mathbf{p}_{\text{ID}_i}^{(v)}. \tag{11}$$

Note that:

1. Since $\hat{\mathbf{q}}$ follows the oral distribution $\mathbf{q}^\star$, it satisfies that for any $i \in [T]$,

$$\mathbf{p}_i^\star > 0 \quad \text{where} \quad \hat{\mathbf{q}}_i \neq 0.$$

Then, we have $D_{\text{KL}}(\hat{\mathbf{q}} \| \mathbf{p}^\star) < \infty$.

2. For any $v' \in \mathcal{V}$, if there exists $i \in [T]$ such that

$$\mathbf{p}_i^{(v')} = 0 \quad \text{where} \quad \hat{\mathbf{q}}_i \neq 0.$$

Then, $D_{\text{KL}}(\hat{\mathbf{q}} \| \mathbf{p}^{(v')}) = \infty$ and $\hat{v} \neq v'$.

Therefore, in this work, we focus only on the nonzero $\mathbf{p}_{\text{ID}_i}^{(v)}$ in (11). Let

$$\mathcal{V}' = \{v \in \mathcal{V} : D_{\text{KL}}(\hat{\mathbf{q}} \| \mathbf{p}^{(v)}) < \infty\}$$

$$= \{v \in \mathcal{V} : \mathbf{p}_{\text{ID}_i}^{(v)} \neq 0, \forall i \in [N_u]\}.$$

Then, the following estimator is equivalent to (11):

$$\hat{v} = \arg\min_{v \in \mathcal{V}'} \sum_{i \in [N_u]} -\log \mathbf{p}_{\mathrm{ID}_i}^{(v)}. \tag{12}$$

**Step 2: Reformulate the problem using random variables and derive concentration bounds (for each $u \in \mathcal{U}$).**

For any $v \in \mathcal{V}'$, define random variables

$$Z_i := -\log \mathbf{p}_{\mathrm{ID}_i}^{(v)} - (-\log \mathbf{p}_{\mathrm{ID}_i}^{\star}) = \log \mathbf{p}_{\mathrm{ID}_i}^{\star} - \log \mathbf{p}_{\mathrm{ID}_i}^{(v)}.$$

By Assumption 3.3, we have $|Z_i| \leq |\log c| =: C$.

Since $z_i$'s are i.i.d. sampled following $\mathbf{p}^{\star}$, we have that

$$\mathbb{E}[Z_i] = D_{\mathrm{KL}}(\mathbf{p}^{\star} \| \mathbf{p}^{(v)}). \tag{13}$$

Recall the estimator from (12). To identify the correct token $v^{\star}$, we need to ensure that

$$\sum_{i \in [N_u]} -\log \mathbf{p}_{\mathrm{ID}_i}^{\star} \leq \sum_{i \in [N_u]} -\log \mathbf{p}_{\mathrm{ID}_i}^{(v)} \implies \sum_{i \in [N_u]} \log \mathbf{p}_{\mathrm{ID}_i}^{\star} - \log \mathbf{p}_{\mathrm{ID}_i}^{(v)} \geq 0$$

$$\implies \sum_{i \in [N_u]} Z_i \geq 0.$$

Given that $|Z_i| \leq C$, by Hoeffding's inequality, for any $t > 0$, we get

$$\mathbb{P}\left( \frac{1}{N_u} \sum_{i \in [N_u]} Z_i - \mathbb{E}[Z_i] \leq -t \right) \leq \exp\left( -\frac{N_u t^2}{2C^2} \right) \implies \mathbb{P}\left( \frac{1}{N_u} \sum_{i \in [N_u]} Z_i \leq \mathbb{E}[Z_i] - t \right) \leq \exp\left( -\frac{N_u t^2}{2C^2} \right).$$

Recall the separation margin $\gamma$ from Definition 3.4 where we have

$$\gamma := \min_{i,j \in [T], i \neq j} D_{\mathrm{KL}}(\mathbf{p}^{(v_i)} \| \mathbf{p}^{(v_j)}).$$

Given (13), we get

$$\mathbb{E}[Z_i] \begin{cases} \geq \gamma > 0 & \text{if } v \in \mathcal{V}', v \neq v^{\star} \\ = 0 & \text{if } v = v^{\star} \end{cases}.$$

Choose $t = \mathbb{E}[Z_i]$. Then we get for any $v \neq v^{\star}$

$$\mathbb{P}\left( \frac{1}{N_u} \sum_{i \in [N_u]} Z_i \leq 0 \right) \leq \exp\left( -\frac{N_u (\mathbb{E}[Z_i])^2}{2C^2} \right) \leq \exp\left( -\frac{N_u \gamma^2}{2C^2} \right).$$

Applying union bound over all $v \in \mathcal{V}' \setminus \{v^{\star}\}$, we get

$$\mathbb{P}(\hat{v} \neq v^{\star}) \leq |\mathcal{V}'| \cdot \exp\left( -\frac{N_u \gamma^2}{2C^2} \right) \leq T \cdot \exp\left( -\frac{N_u \gamma^2}{2C^2} \right). \tag{14}$$

**Step 3: Apply a union bound over all $u \in \mathcal{U}$.**

So far, we have established the probability of successfully retrieving $v^{\star}(u)$ for each $u \in \mathcal{U}$. We now consider the event that retrieval is successful simultaneously for all $u \in \mathcal{U}$.

Applying Lemma A.1, we have that with probability at least $1 - \exp(-N/8m)$ (via choosing $\varepsilon = 0.5$),

$$N_u \geq N/2m.$$

Combining with (14) returns that

$$\mathbb{P}(\hat{v} \neq v^\star) \leq T \cdot \exp\left(-\frac{N\gamma^2}{4mC^2}\right) + \exp\left(-\frac{N}{8m}\right)$$

$$\leq 2T \cdot \exp\left(-\frac{N\min\{\gamma^2/C^2, 1\}}{8m}\right)$$

Applying another union bound over all $u \in \mathcal{U}$ ($|\mathcal{U}| = m$), we have that

$$\mathbb{P}\left(\exists u \in \mathcal{U} \; : \; \hat{v}(u) \neq v^\star(u)\right) \leq 2mT \cdot \exp\left(-\frac{N \cdot \min\{\gamma^2/C^2, 1\}}{8m}\right).$$

Equivalently, with probability at least $1 - \delta$,

$$\forall u \in \mathcal{U} \; : \quad v^\star(u) = \hat{v}(u)$$

when

$$N \geq \frac{1}{\min\{\gamma^2/C^2, 1\}} \cdot 8m \log \frac{2mT}{\delta}.$$

Setting $C = |\log c|$ completes the proof.

$\square$

## A.3. Proof of Theorem 3.7

*Proof.* Recall from Definition 2.1 and Assumption 3.6. For any $u \in \mathcal{U}$, we have

$$\mathbf{q}^{(u)} = f(\mathbf{E}^\top \boldsymbol{\alpha}^{(u)}) \quad \text{where} \quad \boldsymbol{\alpha}^{(u)} \in \mathcal{A} \quad \text{and} \quad \mathcal{A} := \{\boldsymbol{\alpha} \in \mathbb{R}^T \; : \; \|\boldsymbol{\alpha}\|_0 \leq s, \; \|\boldsymbol{\alpha}\|_2 \leq B\}.$$

Following (7), let

$$N_u = \sum_{i \in [N]} \sum_{j \in [t_i]} \mathbf{1}\{x_j^{(i)} = u\},$$

and the empirical distribution $\hat{\mathbf{q}}^{(u)} \in \Delta(\mathcal{V})$ from (4)

$$\hat{\mathbf{q}}_k^{(u)} = \frac{\sum_{i \in [N]} \sum_{j \in [t_i]} \mathbf{1}\{x_j^{(i)} = u, x_{j+1}^{(i)} = v_k\}}{\sum_{i \in [N]} \sum_{j \in [t_i]} \mathbf{1}\{x_j^{(i)} = u\}} \quad \text{for } k \in [T]$$

where $\sum_{k \in [T]} \hat{\mathbf{q}}_k^{(u)} = 1$.

**Step 1: Define the optimization objective and the algorithm for finding the optimal $\hat{v}(u)$ (for each $u \in \mathcal{U}$).**

Fix any $u \in \mathcal{U}$. Similar to the proof of Theorem 3.5, let the samples corresponding to $u$ be $\{z_i\}_{i=1}^{N_u}$, which are i.i.d. from $\mathbf{q}^{(u)}$. Additionally, let $\{\text{ID}_i\}$ be the associated token ID's. For any $\boldsymbol{\alpha} \in \mathcal{A}$, define

$$\mathbf{q}^{(\alpha)} := f(\mathbf{E}^\top \boldsymbol{\alpha})$$

For simplicity, let $\hat{\mathbf{q}} = \hat{\mathbf{q}}^{(u)}$, $\mathbf{q}^\star = \mathbf{q}^{(u)}$, $\hat{\boldsymbol{\alpha}} = \hat{\boldsymbol{\alpha}}^{(u)}$ and $\boldsymbol{\alpha}^\star = \boldsymbol{\alpha}^{(u)}$.

We consider the KL estimator (as in (5))

$$\hat{\boldsymbol{\alpha}} \in \arg\min_{\boldsymbol{\alpha} \in \mathcal{A}} D_{\text{KL}}\left(\hat{\mathbf{q}} \,\|\, \mathbf{q}^{(\alpha)}\right),$$

where we have,

$$\arg\min_{\alpha\in\mathcal{A}} D_{\mathrm{KL}}\left(\hat{\mathbf{q}} \parallel \mathbf{q}^{(\alpha)}\right) = \arg\min_{\alpha\in\mathcal{A}} \sum_{i\in[T]} \hat{\mathbf{q}}_i \log \frac{\hat{\mathbf{q}}_i}{\mathbf{q}_i^{(\alpha)}}$$

$$= \arg\min_{\alpha\in\mathcal{A}} \left( \underbrace{\sum_{i\in[T]} \hat{\mathbf{q}}_i \log \hat{\mathbf{q}}_i}_{\text{constant}} - \sum_{i\in[T]} \hat{\mathbf{q}}_i \log \mathbf{q}_i^{(\alpha)} \right)$$

$$= \arg\min_{\alpha\in\mathcal{A}} \sum_{i\in[T]} -\hat{\mathbf{q}}_i \log \mathbf{q}_i^{(\alpha)}$$

$$= \arg\min_{\alpha\in\mathcal{A}} \sum_{i\in[N_u]} - \log \mathbf{q}_{\mathrm{ID}_i}^{(\alpha)}.$$

Then we can rewrite the estimator via

$$\hat{\alpha} \in \arg\min_{\alpha\in\mathcal{A}} \sum_{i\in[N_u]} - \log \mathbf{q}_{\mathrm{ID}_i}^{(\alpha)}. \tag{15}$$

Following the similar analysis as in the proof of Theorem 3.5, for some $\alpha' \in \mathcal{A}$, if there exists $k \in [T]$ such that

$$\mathbf{q}_k^{(\alpha')} = 0 \quad \text{where} \quad \hat{\mathbf{q}}_k \neq 0,$$

then $\sum_{i\in[N_u]} - \log \mathbf{q}_{\mathrm{ID}_i}^{(\alpha')} = \infty$ and $\hat{\alpha} \neq \alpha'$.

Therefore, in this work, we focus only on the nonzero $\mathbf{q}_{\mathrm{ID}_i}^{(\alpha)}$ in (15). Let

$$\mathcal{A}' = \{\alpha \in \mathcal{A} \; : \; \mathbf{q}_{\mathrm{ID}_i}^{(\alpha)} \neq 0, \forall i \in [N_u]\}.$$

Then, the following estimator is equivalent to (15):

$$\hat{\alpha} \in \arg\min_{\alpha\in\mathcal{A}'} \sum_{i\in[N_u]} - \log \mathbf{q}_{\mathrm{ID}_i}^{(\alpha)}. \tag{16}$$

**Step 2: Reformulate the problem using random variables and derive concentration bounds (for each $u \in \mathcal{U}$).**

For any $\alpha \in \mathcal{A}'$, define the empirical and population losses are as follows:

$$\hat{\mathcal{L}}(\alpha) := \frac{1}{N_u} \sum_{i\in[N_u]} - \log \mathbf{q}_{\mathrm{ID}_i}^{(\alpha)} \quad \text{and} \quad \mathcal{L}(\alpha) := \mathbb{E}_{v_i \sim \mathbf{q}^\star} - \log \mathbf{q}_i^{(\alpha)}.$$

Then we have

$$\mathbb{E}[\hat{\mathcal{L}}(\alpha)] = \mathcal{L}(\alpha) \quad \text{and} \quad \mathcal{L}(\alpha) - \mathcal{L}(\alpha^\star) = D_{\mathrm{KL}}\left(\mathbf{q}^\star \parallel \mathbf{q}^{(\alpha)}\right). \tag{17}$$

We first bound

$$\mathcal{L}(\hat{\alpha}) - \mathcal{L}(\alpha^\star) = D_{\mathrm{KL}}(\mathbf{q}^\star \parallel \mathbf{q}^{(\hat{\alpha})})$$

By the optimality of $\hat{\alpha}$ in (16), we have that

$$\hat{\mathcal{L}}(\hat{\alpha}) \leq \hat{\mathcal{L}}(\alpha^\star).$$

Hence,

$$\mathcal{L}(\hat{\alpha}) - \mathcal{L}(\alpha^\star) = \left(\mathcal{L}(\hat{\alpha}) - \hat{\mathcal{L}}(\hat{\alpha})\right) + \left(\hat{\mathcal{L}}(\hat{\alpha}) - \hat{\mathcal{L}}(\alpha^\star)\right) + \left(\hat{\mathcal{L}}(\alpha^\star) - \mathcal{L}(\alpha^\star)\right)$$

$$\leq 2 \max_{\alpha\in\mathcal{A}'} \left|\hat{\mathcal{L}}(\alpha) - \mathcal{L}(\alpha)\right|. \tag{18}$$

Therefore, it suffices to control the uniform deviation of $\hat{\mathcal{L}}$ from $\mathcal{L}$ over $\mathcal{A}'$.

By Assumption 3.3, we have $0 \leq -\log \mathbf{q}_{\mathrm{ID}_i}^{(\alpha)} \leq |\log c| =: C$. Then, applying Hoeffding's inequality, for $\alpha \in \mathcal{A}'$, we have that

$$\mathbb{P}\left(\left|\hat{\mathcal{L}}(\alpha) - \mathcal{L}(\alpha)\right| \geq t\right) \leq 2 \exp\left(-\frac{2t^2 N_u}{C^2}\right). \tag{19}$$

Next, since for any $x, y \in [c, 1]$

$$|\log x - \log y| \leq \frac{1}{c}|x - y|.$$

Then we have that

$$\left|\hat{\mathcal{L}}(\alpha^\star) - \hat{\mathcal{L}}(\hat{\alpha})\right| \leq \frac{1}{c}\|\mathbf{q}^\star - \mathbf{q}^{(\hat{\alpha})}\|_1.$$

Similarly,

$$\left|\mathcal{L}(\alpha^\star) - \mathcal{L}(\hat{\alpha})\right| \leq \frac{1}{c}\|\mathbf{q}^\star - \mathbf{q}^{(\hat{\alpha})}\|_1.$$

Next, since $f$ is $L$-Lipschitz from $(\mathbb{R}^d, \|\cdot\|_2)$ to $(\mathbb{R}^T, \|\cdot\|_1)$. Then

$$\|\mathbf{q}^\star - \mathbf{q}^{(\hat{\alpha})}\|_1 = \|f(\mathbf{E}^\top \alpha^\star) - f(\mathbf{E}^\top \hat{\alpha})\|_1 \leq L\|\mathbf{E}^\top \alpha^\star - \mathbf{E}^\top \hat{\alpha}\|_2 \leq L \cdot \sigma_{\max}(\mathbf{E})\|\alpha^\star - \hat{\alpha}\|_2,$$

where $\sigma_{\max}(\mathbf{E})$ returns the maximal singular value of matrix $\mathbf{E}$. Combining, we obtain for any $\alpha, \alpha' \in \mathcal{A}'$

$$\left|\hat{\mathcal{L}}(\alpha) - \hat{\mathcal{L}}(\alpha')\right| \vee \left|\mathcal{L}(\alpha) - \mathcal{L}(\alpha')\right| \leq \underbrace{\frac{L\sigma_{\max}(\mathbf{E})}{c}}_{=:\tilde{L}}\|\alpha - \alpha'\|_2.$$

Let $\{\alpha^{(k)}\}_{k=1}^M$ be the $\eta$-covering of $\mathcal{A}'$ in $\ell_2$ where $M := \mathcal{N}(\mathcal{A}', \|\cdot\|_2, \eta)$. Following (Vershynin, 2018), $M$ is bounded by a standard covering bound for $s$-sparse vectors in an $\ell_2$-ball of radius $B$, given by

$$M \leq \binom{T}{s}\left(\frac{3B}{\eta}\right)^s \leq \left(\frac{eT}{s}\right)^s\left(\frac{3B}{\eta}\right)^s \leq \left(\frac{9BT}{s\eta}\right)^s.$$

Note that for any $\alpha \in \mathcal{A}'$, we can pick $\alpha^{(k)}$ such that $\|\alpha - \alpha^{(k)}\| \leq \eta$. Then,

$$\left|\hat{\mathcal{L}}(\alpha) - \mathcal{L}(\alpha)\right| \leq \left|\hat{\mathcal{L}}(\alpha) - \hat{\mathcal{L}}(\alpha^{(k)})\right| + \left|\hat{\mathcal{L}}(\alpha^{(k)}) - \mathcal{L}(\alpha^{(k)})\right| + \left|\mathcal{L}(\alpha^{(k)}) - \mathcal{L}(\alpha)\right|$$
$$\leq \left|\hat{\mathcal{L}}(\alpha^{(k)}) - \mathcal{L}(\alpha^{(k)})\right| + 2\tilde{L}\eta.$$

Recall (19). Taking a union bound gives

$$\mathbb{P}\left(\sup_{\alpha \in \mathcal{A}'}\left|\hat{\mathcal{L}}(\alpha) - \mathcal{L}(\alpha)\right| \geq t + 2\tilde{L}\eta\right) \leq 2M \exp\left(-\frac{2t^2 N_u}{C^2}\right)$$

which returns (applying (18))

$$\mathbb{P}\left(\mathcal{L}(\hat{\alpha}) - \mathcal{L}(\alpha^\star) \geq 2t + 4\tilde{L}\eta\right) \leq 2M \exp\left(-\frac{2t^2 N_u}{C^2}\right). \tag{20}$$

### Step 3: Apply a union bound over all $u \in \mathcal{U}$.

So far, we have derived an upper bound for each $u \in \mathcal{U}$. We now consider the bound holds uniformly over all $u \in \mathcal{U}$.

Similarly, applying Lemma A.1, we have that with probability at least $1 - \exp(-N/8m)$ (via choosing $\varepsilon = 0.5$),

$$N_u \geq N/2m.$$

Combining with (20) returns that

$$\mathbb{P}\left(\mathcal{L}(\hat{\alpha}) - \mathcal{L}(\alpha^\star) \geq 2t + 4\tilde{L}\eta\right) \leq 2M \exp\left(-\frac{t^2 N}{mC^2}\right) + \exp\left(-\frac{N}{8m}\right).$$

Recalling the estimation risk from (6) and applying (17), we get

$$\mathcal{R}(\{\hat{\boldsymbol{\alpha}}^{(u)}\}_{u \in \mathcal{U}}) = \max_{u \in \mathcal{U}} D_{\text{KL}}(\mathbf{q}^{(u)} \| f(\mathbf{E}^\top \hat{\boldsymbol{\alpha}}^{(u)}))$$

$$= \max_{u \in \mathcal{U}} D_{\text{KL}}(\mathbf{q}^{(u)} \| \mathbf{q}^{(\hat{\alpha}^{(u)})})$$

$$= \max_{u \in \mathcal{U}} \mathcal{L}(\hat{\boldsymbol{\alpha}}^{(u)}) - \mathcal{L}(\boldsymbol{\alpha}^{(u)}).$$

Applying another union bound over all $u \in \mathcal{U}$ ($|\mathcal{U}| = m$), we have that

$$\mathbb{P}\left(\mathcal{R}(\{\hat{\boldsymbol{\alpha}}^{(u)}\}_{u \in \mathcal{U}}) \geq 2t + 4\tilde{L}\eta\right) \leq 2mM \exp\left(-\frac{t^2 N}{mC^2}\right) + m \exp\left(-\frac{N}{8m}\right)$$

$$\leq 2m\left(\frac{9BT}{s\eta}\right)^s \exp\left(-\frac{t^2 N}{mC^2}\right) + m \exp\left(-\frac{N}{8m}\right).$$

Equivalently,

$$\mathbb{P}\left(\mathcal{R}(\{\hat{\boldsymbol{\alpha}}^{(u)}\}_{u \in \mathcal{U}}) \geq t + \varepsilon\right) \leq 2m\left(\frac{36BTL\sigma_{\max}(\mathbf{E})}{cs\varepsilon}\right)^s \exp\left(-\frac{t^2 N}{4m \log^2 c}\right) + m \exp\left(-\frac{N}{8m}\right).$$

1. Setting $m \exp\left(-\frac{N}{8m}\right) \leq \frac{\delta}{2}$ returns

$$N \geq 8m \log \frac{2m}{\delta}.$$

2. Setting

$$2m\left(\frac{36BTL\sigma_{\max}(\mathbf{E})}{cs\varepsilon}\right)^s \exp\left(-\frac{t^2 N}{4m \log^2 c}\right) = \frac{\delta}{2}$$

returns

$$t = \sqrt{\frac{4m \log^2 c}{N}\left(s \log \frac{36BTL\sigma_{\max}(\mathbf{E})}{cs\varepsilon} + \log \frac{4m}{\delta}\right)}.$$

Combining gets, when

$$N \geq O\left(m \log(m/\delta)\right),$$

we have that with probability at least $1 - \delta$,

$$\mathcal{R}(\{\hat{\boldsymbol{\alpha}}^{(u)}\}_{u \in \mathcal{U}}) \leq \min_{\varepsilon > 0}\left\{\varepsilon + O\left(\sqrt{\frac{m \log^2 c}{N}\left(\log \frac{m}{\delta} + s \log \frac{T}{cs\varepsilon}\right)}\right)\right\}.$$

It completes the proof.

$\square$

## B. Experimental Setup and Implementation Details

### B.1. Synthetic Sentence Generation

We construct a synthetic corpus using the **Phi-3.5 Mini Instruct** language model to study embedding-based adaptation under controlled vocabulary shifts. The model is used both to generate coherent natural-language sentences and to introduce controlled lexical perturbations through synthetic, non-sense word usage.

We define several semantic domains, including *food*, *nature*, *office*, *household*, and *transportation*. This setup is closely related to prior work on controlled distribution shifts and lexical perturbations in language models (Ben-David et al., 2010; Ribeiro et al., 2020), and follows recent uses of synthetic data for analyzing robustness and adaptation in large language models (Gururangan et al., 2020; Muennighoff et al., 2023). For each domain, we curate a list of real-world vocabulary items. In this appendix, we describe the *food* domain in detail; all other domains follow the same procedure.

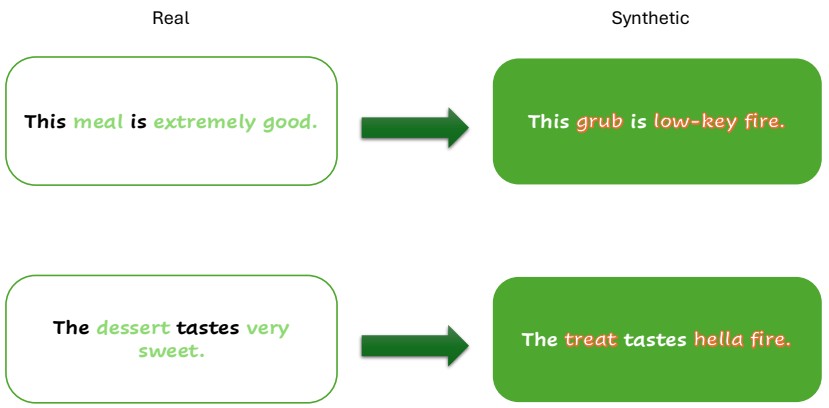

*Figure 3.* Illustration of synthetic sentence creation from real sentences.

From the food-domain vocabulary, we select a subset of 100 real words (e.g., *broccoli, cumin, lentils, apple, basil*). Using **Phi-3.5 Mini Instruct**, we prompt the model to generate natural, declarative sentences such that each sentence contains at least one word from the selected vocabulary list. The model is instructed to output only complete sentences, without explanations, lists, or formatting artifacts.

Examples of generated sentences from the food domain are shown below.

Figure 3 illustrates the process of creating synthetic sentences from real sentences. The left column shows original sentences sampled from a real domain (e.g., food), and the right column shows the corresponding synthetic sentences.

In this diagram, we highlight the key steps:

1. **Selection of Real Tokens:** Since the corpus is generated using a fixed set of 100 real food-domain words, we treat these 100 words as the *target vocabulary* for synthetic replacement. Thus, for each real sentence, we identify occurrences of any target word from this set (e.g., meal, dessert, **extremely good**, **very sweet**) that will be transformed in the synthetic version.

2. **Replacement with Synthetic Tokens:** Each selected real token is replaced with a semantically meaningless synthetic token, while the corresponding descriptive phrase is also highlighted. For instance, "This meal is **extremely good**" becomes "This grub is **low-key fire**," and "The dessert tastes **very sweet**" becomes "The treat tastes **hella fire**."

3. **Context Preservation:** The replacement ensures that only the chosen lexical items and descriptive phrases change, while sentence syntax and surrounding context remain intact, maintaining comparability between real and synthetic sentences.

4. **Synthetic Dataset Construction:** Repeating this process over multiple sentences generates a large set of real-synthetic sentence pairs. These pairs are used for controlled experiments to evaluate embedding adaptation, vocabulary learning, and catastrophic forgetting.

To construct synthetic sentence pairs, we define a separate list of semantically meaningless tokens (e.g., *Blorpt, Snazzit, Quorfle, Trimplix, Whimzor*). These fake words do not appear in the pretrained vocabulary and are introduced solely to induce controlled lexical shifts.

### B.1.1. LoRA IMPLEMENTATION

We include LoRA as a standard parameter-efficient fine-tuning (PEFT) baseline for learning the synthetic vocabulary task. LoRA freezes the pretrained backbone and injects trainable low-rank matrices into selected transformer modules, enabling adaptation with a small number of additional parameters. In our implementation, LoRA adapters are inserted into the attention projection layers, and only the LoRA parameters are optimized on the synthetic-token training set. We use rank $r = 8$, scaling $\alpha = 32$, dropout 0.05, and no bias parameters.

*Table 4.* Trainable parameter counts for LoRA (rank $r = 8$, $\alpha = 32$), soft prompt tuning (prompt length = 20), and embedding tuning when adding 100 new tokens (reporting only the effective trainable embedding rows).

| Model | Total Params | LoRA Params | Soft Prompt Params | Embed Params (+100 toks) |
|---|---|---|---|---|
| Phi-3.5-mini-Instruct | 3.80B | 2,883,584 | 61,440 | 614,400 |
| Llama-3.2-1B | 1.00B | 1,703,936 | 40,960 | 204,800 |
| Llama-3.1-3B | 3.21B | 4,587,520 | 61,440 | 307,200 |
| Llama-3.1-8B | 8.03B | 6,815,744 | 81,920 | 819,200 |
| Qwen2.5-1.5B | 1.54B | 2,179,072 | 30,720 | 153,600 |
| Qwen2.5-3B | 3.09B | 3,686,400 | 40,960 | 204,800 |
| Qwen2.5-7B | 7.61B | 5,046,272 | 71,680 | 358,400 |

*Table 5.* Arabic performance and English forgetting across methods for different models. Lower Arabic loss is better. Forgetting is measured as the increase in English loss after adaptation; lower is better.

| Model | Performance (Arabic loss) | | | | Base English loss | Forgetting (English) | | | |
|---|---|---|---|---|---|---|---|---|---|
| | FFT | LoRA | PT | ET | | FFT | LoRA | PT | ET |
| Llama3.2-1B | 5.27 | 3.65 | 6.05 | **3.47** | 2.18 | +13.59 | +0.35 | +0.42 | **0.0001** |
| Llama3.2-3B | 7.43 | 3.57 | 5.62 | **3.15** | 1.91 | +14.22 | +0.39 | +0.37 | **0.0003** |
| Qwen2.5-3B | 7.95 | 3.95 | 4.34 | **2.82** | 2.01 | +11.95 | +0.46 | +0.06 | **0.00** |
| Qwen2.5-7B | 6.18 | 3.90 | 4.59 | **2.76** | 1.90 | +19.19 | +0.18 | +0.09 | **-0.15** |

**Results and Forgetting.** Tables 2 summarize LoRA's performance. LoRA often achieves competitive convergence on the synthetic vocabulary task, e.g., on LLaMA-3.2-1B its test loss is close to embedding tuning (ET) and substantially better than full fine-tuning (FFT). However, LoRA still induces non-negligible forgetting on WikiText, with loss increases that can remain substantial in some backbones (e.g., Phi-3.5 Mini). Overall, LoRA reduces forgetting compared to FFT but is consistently worse than ET, which achieves zero forgetting across all models.

B.1.2. EMBEDDING TUNING IMPLEMENTATION

To study whether pretrained LMs can acquire novel synthetic vocabulary without disrupting existing linguistic knowledge, we perform *embedding tuning* (ET), where only the parameters corresponding to newly introduced tokens are updated. For each backbone model (LLaMA-3.2-1B, LLaMA-3.1-8B, Phi-3.5-mini-Instruct, and Qwen2.5-3B-Instruct), we extend the tokenizer with 100 semantically meaningless synthetic tokens (e.g., `Quib`, `Flem`, `Snix`, ...) using `additional_special_tokens`, and resize the embedding matrix accordingly.

**Results and Forgetting.** ET yields the most parameter-efficient adaptation: updating only the synthetic token rows corresponds to $2 \times 100 \times d$ trainable parameters (embedding + LM head), which is substantially smaller than LoRA and full fine-tuning (see Table 4). Empirically, ET achieves consistently low synthetic test loss while also preserving pretrained capabilities, exhibiting near-zero (often zero) forgetting on WikiText across all model scales (Table 2)

Table 4 summarizes the trainable parameter budgets of different PEFT methods across a range of model families and scales. LoRA introduces approximately 1.7M–6.8M trainable parameters, which is substantially larger than soft prompt tuning (~31K–82K) but still far smaller than full fine-tuning. In contrast, embedding tuning scales linearly with the number of added tokens and the model hidden dimension, requiring 153K–819K effective trainable parameters when adding +100 synthetic tokens. Overall, soft prompt tuning has the smallest trainable footprint, LoRA provides a moderate parameter budget, and embedding tuning lies in between while remaining highly parameter-efficient relative to full fine-tuning.

**B.2. Cross-lingual Vocabulary Expansion**

Table 3 reports cross-lingual vocabulary expansion results on Qwen2.5-3B for Spanish, German, and Arabic. Table 5 further reports Arabic results across multiple backbone models. In all cases, lower target-language loss indicates better adaptation, while forgetting is measured as the increase in English loss after adaptation. Across all the settings, the results suggest that embedding tuning remains effective even under realistic experimental scenarios.

# C. Extended Related Work

## C.1. Continual Learning and Forgetting in LLMs

Catastrophic forgetting remains a fundamental challenge when neural networks are updated sequentially. Classical approaches such as Elastic Weight Consolidation (Kirkpatrick et al., 2017) and memory-based methods like GEM (Lopez-Paz & Ranzato, 2017) and A-GEM (Chaudhry et al., 2019) mitigate forgetting in small-scale networks, but often fail on large language models (LLMs) due to scale and complexity (Wu et al., 2024). Comprehensive surveys by (De Lange et al., 2021) and more recently (Shi et al., 2025) categorize these strategies, yet scaling them effectively remains an open problem.

Parameter-efficient fine-tuning (PEFT) methods, including adapters and LoRA, were designed to reduce forgetting by isolating task-specific updates (Hu et al., 2021; Pfeiffer et al., 2021). However, recent work indicates that even PEFT methods can suffer from "gradual forgetting" before reaching a catastrophic tipping point when updates accumulate (Gupta et al., 2024; Bafghi et al., 2024). Retrieval-augmented methods leverage external memory to partially preserve knowledge (Lewis et al., 2020; Muennighoff et al., 2023), but fine-tuning for RAG capabilities can paradoxically induce distribution shifts that degrade general instruction-following abilities (Huang et al., 2025).

Furthermore, targeted model editing approaches, often viewed as precise surgical updates, have been shown to cause abrupt "disabling edits" where a single update cripples general model function (Meng et al., 2022; Gupta et al., 2024). These findings highlight the critical need for principled continual update strategies that go beyond simple parameter isolation or external memory.

## C.2. Embedding and Parameter-Efficient Tuning

Parameter-efficient fine-tuning (PEFT) techniques update only a small subset of model parameters, enabling the integration of new knowledge while preserving existing model capabilities. These approaches include embedding updates, prefix-tuning (Li & Liang, 2021) and P-tuning v2 (Liu et al., 2022), LoRA (Hu et al., 2021), adapters, including multi-task-aware AdapterFusion (Pfeiffer et al., 2021), and BitFit (Ben Zaken et al., 2022). Empirical studies have demonstrated that such methods effectively incorporate new knowledge with reduced risk of catastrophic forgetting compared to full fine-tuning (Meng et al., 2022; Muennighoff et al., 2023; Lester et al., 2021; He et al., 2020). Recent advances have further refined these techniques: Xiong & Xie (2025) introduced Orthogonal Projection LoRA (OPLoRA) to mathematically guarantee non-interference with pretrained weights, while Chen et al. (2024) proposed the Momentum-Filtered Optimizer (MoFO) to mitigate forgetting via sparse, momentum-guided updates without requiring pretraining data. Despite these practical successes, a formal theoretical understanding of exactly why specific sparse update patterns, like embedding tuning, prevent forgetting remains limited (Meng et al., 2022; Jiang et al., 2024).

## C.3. Connection to Sparse Representation Learning

Our token-to-dictionary mapping framework connects naturally to sparse representation learning and dictionary learning theory (Olshausen & Field, 1997; Elad, 2010). In our formulation, learning a new token $u \in U$ corresponds to finding a sparse representation $\alpha(u) \in \mathbb{R}^T$ such that the new token's embedding can be expressed as a sparse linear combination of existing token embeddings. This perspective enables us to apply classical sparse coding theory: the sample complexity bounds in Theorem 3.7 directly reflect the cost of learning sparse combinations, with complexity scaling in the sparsity parameter $s$ rather than the vocabulary size $T$. This connection also suggests that our framework could benefit from advances in structured sparsity and learned dictionary methods.

## C.4. Domain Adaptation and Vocabulary Expansion

Domain adaptation enables models to specialize in new domains, but can lead to degradation when internal representations drift (Ben-David et al., 2010; Gururangan et al., 2020; Muennighoff et al., 2023). Full-parameter fine-tuning amplifies this risk (Muennighoff et al., 2023). Frameworks like AdapterHub facilitate the modular reuse of adapters across tasks, empirically improving adaptability while reducing interference (Pfeiffer et al., 2021). Recent approaches have focused on explicitly expanding the model's vocabulary to better represent domain-specific concepts: Liu et al. (2024) demonstrated that adaptively selecting a subset of high-value domain tokens (VEGAD) outperforms full vocabulary expansion, while Wei et al. (2024) successfully scaled vision vocabularies in multimodal models without overwriting original knowledge. Similarly, Yamaguchi et al. (2025b) showed that strategic vocabulary expansion can be achieved with minimal data (~0.01GB), enabling low-resource adaptation. Our work complements these methods by introducing a principled framework for

vocabulary expansion via embedding updates, which update only new token embeddings, thereby minimizing interference with existing token representations and supporting stable continual updates.

### C.5. Knowledge Retention in LLMs

Recent studies emphasize maintaining prior knowledge while adapting LLMs. Model editing approaches (Meng et al., 2022) and modular fine-tuning strategies (Hu et al., 2021; He et al., 2020) demonstrate empirical retention of prior capabilities. Sparse or selective update strategies have further been shown to mitigate forgetting in LLMs by isolating critical parameters (Chen et al., 2025). For instance, Source-Shielded Updates (SSU) explicitly protect pre-trained weights during adaptation, reducing forgetting on source tasks to under 4% (Yamaguchi et al., 2025a). However, few existing methods provide formal guarantees of knowledge retention, particularly when introducing new tokens or domain-specific vocabulary. Our approach differs fundamentally: rather than relying on regularization or external memory mechanisms, we guarantee preservation *by construction*. The Markovian formulation ensures that embedding updates leave token transition probabilities invariant, enabling formal proofs of zero forgetting on the original vocabulary. While recent work such as Orthogonal Projection LoRA (OPLoRA) offers mathematical guarantees for preserving top-$k$ singular values during fine-tuning (Xiong & Xie, 2025), theoretical frameworks specifically addressing vocabulary expansion remain scarce. Our framework addresses this gap by formally modeling language generation as a Markov process over tokens, under which updating embeddings alone preserves the original token-level transition structure by construction.

Recent work on model editing (Meng et al., 2022; Gupta et al., 2024) targets fact-level modifications, whereas our framework provides principled vocabulary expansion with formal sample complexity bounds. The token-to-dictionary mapping could extend model editing to handle concurrent updates more robustly. Nested Learning and our work share the goal of continual adaptation with minimal forgetting, but they operate at different levels. Nested Learning is a general optimization and architectural framework for continual learning, whereas our paper studies a specific post-training setting for autoregressive language models: adding new vocabulary as state-space expansion in a Markov process. Our contribution is a token-level formulation with sample-complexity guarantees for token-to-dictionary mappings realized through embedding tuning. We therefore view the two directions as complementary rather than competing.

