# OpenReview forum: "Memory as a Markov Matrix: Sample Efficient Knowledge Expansion via Token-to-Dictionary Mapping"
_ICML.cc/2026/Conference — ICML 2026 regular_

### Official Review · Reviewer_Tg5H · 2026-03-02

**Soundness:** 3
**Presentation:** 3
**Significance:** 3
**Originality:** 3
**Overall Recommendation:** 4
**Confidence:** 2

**Summary:**

This paper addresses catastrophic forgetting in LLMs during knowledge expansion by innovatively modeling text generation as a Markov process over tokens. The authors propose a token-to-dictionary mapping strategy, practically implemented via embedding tuning for new tokens. By preserving original transition probabilities, this approach mathematically mitigates the forgetting of prior knowledge. Finally, the paper designs experiments targeting this problem, including new arithmetic operations and vocabulary synthesis tasks. These experiments validate the method's sample efficiency and anti-forgetting properties in low-data regimes.

**Compliance With Llm Reviewing Policy:**

Affirmed.

**Final Justification:**

Thank you for the rebuttal. I appreciate the theoretical clarifications you provided, yet my concerns about the practical limitations in spontaneous generation and factual updates still remain. I will keep my current score accordingly.

**Key Questions For Authors:**

**Q1:** If the transition probability from existing tokens to new tokens is strictly restricted to zero, how can the model spontaneously generate this new knowledge or these new tokens without explicit prompt cues? If the model cannot actively recall new words in open-ended question-answering, what are the actual practical application scenarios for this kind of "knowledge expansion"? Could the authors demonstrate how the forgetting rate changes when this assumption is relaxed?

**Q2:** The current experiments are primarily based on highly idealized synthetic tasks (e.g., mapping arithmetic operators and synthesizing or replacing nonsense words). How can this method be applied to more complex, real-world corpora? For example, can this method be applied to TiC-LM [1]? If so, how should it be implemented? If not, what are the underlying reasons?

**Q3:** In the real world, some knowledge updates may not require adding new tokens (for instance, Twitter changing its name to X). How does the method proposed in this paper address the catastrophic forgetting problem when only the relationships among existing tokens need to be updated?


[1] Li J, Armandpour M, Mirzadeh S I, et al. TiC-LM: A web-scale benchmark for time-continual LLM pretraining[C]//Proceedings of the 63rd Annual Meeting of the Association for Computational Linguistics (Volume 1: Long Papers). 2025: 32231-32273.

**Limitations:**

yes

**Strengths And Weaknesses:**

### Strengths
* **Innovative Modeling:** This method innovatively models the autoregressive generation process of LLMs as a Markov transition matrix among tokens. By expanding the Markov state space without altering the original state transition probabilities, it ensures that there is no forgetting when learning new words/tokens.
* **Theoretical Rigor:** The extension of new knowledge is achieved through a Token-to-Dictionary Mapping strategy. The authors provide mathematical explanations and derive sample complexity bounds for learning new tokens under both one-to-one mapping and sparse combination scenarios. This successfully demonstrates that the cost of learning a new token is independent of the model's parameter count and vocabulary size.
* **Targeted Experimental Design:** To validate the sample efficiency and anti-forgetting performance of the method, the authors specifically designed an arithmetic operator task (to test logical mapping) and a synthetic vocabulary task (to test contextual semantic mapping).
* **Overall Significance:** Overall, this paper identifies the pain points of continual learning in existing LLMs and provides a low-cost, highly efficient vocabulary expansion strategy. This holds significant value for the anti-forgetting knowledge expansion of LLMs.
### Weaknesses
* **Restrictive Zero-Transition Assumption (Existing to New):** Regarding the assumption that there are no transitions from existing tokens to new tokens ($p_{\tilde{\theta}}(u|v)=0$), does this imply that the model can only output content containing new tokens if the input prompt explicitly includes tokens from $\mathcal{U}$? The constraints of this assumption appear overly restrictive.
* **Restrictive Zero-Transition Assumption (Among New Tokens):** Regarding the assumption of no transitions among new tokens ($p_{\tilde{\theta}}(u_i|u_j)=0$), if the paths between new tokens are severed, wouldn't the model be unable to generate coherent proper phrases or segments containing multiple new tokens? Furthermore, If new tokens are introduced one by one in sequence for embedding tuning, will this lead to excessive inefficiency in engineering implementation?
* **Experimental Limitations:** Regarding the empirical section, although it validates the authors' claims, the task setups are overly simplistic, and the baseline comparisons are not comprehensive enough.

---

> ### Author Rebuttal · Authors · 2026-03-31
>
> Thank you for your thoughtful review and for recognizing the novelty of our work. We address your concerns below.
>
> **Response to W1/W2:**
> We kindly refer the reviewer to our responses to **Reviewer 8aW5, W1** for more clarification, and to **Reviewer UvGZ's, W2** for additional experiments. In brief, these assumptions are introduced for theoretical analysis only and are not used in the experiments. Additionally, when allowing transitions from existing to new tokens and/or among new tokens, similar sample complexity guarantees can be obtained, though this requires additional assumptions on the model and transition structure. We will add a subsection in our manuscript to discuss these extensions.
>
> **Response to W3:**
> Most existing datasets are general and do not provide the controlled structure, e.g., knowledge expansion, needed to validate our theoretical finding. That is why we constructed our own synthetic dataset, which allows us to precisely evaluate the sparse token-to-dictionary mapping regime studied in the paper. We also emphasize that, although synthetic, the data is constructed from real language sentences. Additionally, following **Reviewer UvGZ** (response to **W2**), we have also added English–Spanish vocabulary-expansion experiments to further demonstrate that our approach extends beyond purely synthetic settings.
>
> **Response to Q1:**
> First, we emphasize that the assumption of no transitions from existing tokens to new tokens is introduced only for theoretical analysis, to enable a clean and well-defined notion of model forgetting. As the reviewer correctly points out, once this assumption is relaxed, precisely quantifying forgetting becomes more challenging.
>
> One natural way to define forgetting in this case is via relative transition probabilities, i.e.,
>
> $$\frac{P(v_i\to v_j)}{\sum_{v_k\in\mathcal{V}}P(v_i\to v_k)}.$$
> Under this definition, introducing new tokens does not cause any forgetting, since the relative ratios $P(v_i\to v_j)/P(v_i\to v_k)$ among existing tokens remain unchanged. However, the absolute transition probabilities $P(v_i\to v_j)$ may shift due to probability mass being redistributed to the newly introduced tokens.
>
> Additionally, in our experiments (see response to **Reviewer UvGZ, W2**), we empirically validate that the forgetting is small, and can even be negative when using our embedding tuning method.
>
>
> **Response to Q2:**
> Our method is most naturally applicable to real-world corpora when the update is primarily lexical.  For TiC-LM, our method is applicable to the new-vocabulary / lexical-expansion part, but not to the full benchmark. TiC-LM involves broader time-continual changes beyond token addition, such as semantic drift, factual updates, and changing relations among existing tokens. Our theory is specifically about adding new tokens while preserving the original vocabulary dynamics, so it does not fully cover those broader updates. Thus, ET is best viewed as a complementary method for lexical expansion, not a complete replacement for full continual pretraining.
> We also include preliminary cross-lingual vocabulary expansion experiments (see **Reviewer UvGZ, W2**), which better align with our setting and demonstrate applicability beyond synthetic tasks.
>
> **Response to Q3:**
> Thank you for this important question. We use the term *knowledge expansion* to specifically refer to adding new tokens, as opposed to modifying relationships among existing tokens. The latter is more closely related to a continual learning setting. One reason we do not study this case in the current work is that quantifying forgetting becomes more challenging when the original transition structure itself changes.
>
> Still, we believe our work provides a novel perspective by connecting embedding tuning with token-to-token mappings, and our empirical results demonstrate its effectiveness in the expansion setting.
>
> One possible way to extend our framework to handle updates among existing tokens is to treat the modified tokens as new tokens. Under this view, and using a relative notion of forgetting (as discussed in our response to **Q1**), similar sample complexity analysis can be applicable. We view this as an interesting direction for future work and will clarify this discussion in the revision.

---

> > ### Author Rebuttal · Reviewer_Tg5H · 2026-04-01
> >
> > Thank you for the rebuttal. While I appreciate the theoretical clarifications, my concerns regarding practical limitations in spontaneous generation and factual updates remain. I will maintain my current score, but strongly urge the authors to explicitly state these boundaries in the revision.

---

> > > ### Author Response · Authors · 2026-04-02
> > >
> > > We thank the reviewer for the continued engagement. However, we would like to respectfully clarify that, as noted in our response to Q1, **we do not impose the zero-transition assumption in our experiments**. That said, our method works broadly in practice.
> > >
> > > - Empirically: The use of synthetic vocabulary in Sec. 5.2 is motivated by the lack of real-world datasets specifically designed for controlled factual updates, as discussed in our **response to W3**. Nevertheless, the constructed data is derived from natural language sentences, ensuring practical relevance. Additionally, following another reviewer’s suggestion (**Reviewer UvGz, W2**), we have included English-to-Spanish expansion experiments to further demonstrate applicability in more realistic settings. *For both experiments, we train the model using the standard next-token prediction objective with cross-entropy loss, without introducing any additional data restrictions.*
> > >
> > > - Theoretically: Importantly, the concerns regarding the zero-transition assumption pertain only to our theoretical analysis. As mentioned in our earlier **response to Q1**, this assumption can be relaxed under appropriate conditions with suitable modifications.
> > >
> > > We hope this clarification addresses the reviewer’s concerns. We will also incorporate a more explicit discussion in the revision.

---

### Official Review · Reviewer_UvGz · 2026-03-13

**Soundness:** 3
**Presentation:** 3
**Significance:** 3
**Originality:** 3
**Overall Recommendation:** 5
**Confidence:** 3

**Summary:**

This paper proposes a framework for vocabulary expansion in LLMs, by modeling autoregressive generation as a Markov process over tokens.  The authors interpret the model’s memory as a Markov transition matrix and frame knowledge expansion as an extension of the state space while preserving original transition dynamics. To integrate new tokens without catastrophic forgetting, they introduce a token-to-dictionary mapping strategy implemented via embedding tuning. Empirical results on arithmetic and synthetic vocabulary tasks support the claims.

**Compliance With Llm Reviewing Policy:**

Affirmed.

**Final Justification:**

The authors addressed part of my concerns during the rebuttal and indicated that they are prepared to incorporate these changes in the revision. I have increased my soundness and confidence scores accordingly. The remaining concern, demonstrating the method’s practicality, is acknowledged in their plan, but no results have been provided yet.

----

**April 11 Update**

The newly added experiment demonstrating real-world cross-lingual transfer between distant language pairs (Arabic–English) provides a positive signal and partially addresses my concern regarding the practicality of the method. I increased my final score conditional on the authors incorporating the following promised revisions (summarized from the rebuttal discussion):

a) **problem framing and literature positioning**

   * a clearer discussion of the relationship between *knowledge expansion* and *vocabulary expansion*.
   * more explicitly positioned within the literature, particularly with respect to [1]–[5].
   * vocabulary expansion is a key aspect of continual learning for large language models. As I understand it, the proposed method operates in a post-training setting, whereas work such as [3] focuses on pretraining. Given that the new experimental setup brings the paper closer in spirit to [3], this distinction should be clearly articulated.

b) **rebuttal experiments and practical relevance**

   * include a dedicated section describing the new experiments introduced during rebuttal. cross-lingual transfer should be presented as a concrete testbed for evaluating the method.
   * detail the new experimental setup (e.g., model choices, metrics) clearly
   * a clearer explanation of why the method is expected to perform better on distant language pairs, potentially drawing on insights from [3]. It may also be useful to note that the method is expected to have limited impact for closely related language pairs. This would improves the paper's impact on NLP community

**Key Questions For Authors:**

Q1. see weakness 2

Q2. can your theory be applied to explain why embedding resetting during pretraining benefits continual learning?

Q3. under your theory, does embedding-tuning only change token-specific information but retain token-token relationship? can you explain if this is true if you are just updating a partial set (S) of the embeddings instead of all the embeddings (A) ? In this case, would not the  relationship between tokens in S and tokens in A change as well?

**Limitations:**

Yes

**Strengths And Weaknesses:**

Strengths

- the topic of LLM continual learning is timely.

- the markov modelling of language is intuitive and novel. While I have personally thought of it a couple of times, this paper is the first time I saw some paper actually spells this out elegantly. The key element here is to see LM as a Markov transition matrix over the token vocabulary, the token embeddings serve as continuous state representations of the underlying Markov process, and then the expansion of knowledge becomes the expansion of the state space.

- the theory kinds of explain why the approach of embedding-only updates may mitigate forgetting


Weakeness

- the terminology is a bit confusing to me. it seems that knowledge expansion here is defined as equal to vocabulary expansion.

- the experiments are limited to synthetic set up. I wonder if it is possible to do a real vocabulary expansion in practice. For example, if I have a LM in English, can I use the proposed method to quickly extend to model to Spanish vocabulary?

- there is some missing related work on vocab expansion. For example, a very relevant line of work is called cross-lingual transfer or cross-lingual vocabulary expansion, where people try to expand a given model's vocabulary to include the vocab from new languages. It could be worthwhile to check out and cite those works properly (e.g.[1]-[5]). While the authors position their contribution as vocabulary expansion, these related work in cross-lingual transfer has studied embedding relearning and reinitialization for expanding multilingual vocab and transfer the model to new languages extensively. A clearer comparison would clarify how the proposed framework differs from or complements these approaches.

[1] Artetxe et al. On the Cross-Lingual Transferability of Monolingual Representations. ACL 2020.

[2] Marchisio et al. Mini-Model Adaptation: Efficiently Extending Pretrained Models to New Languages via Aligned Shallow Training. Findings of ACL 2023.

[3] Chen et al. Improving Language Plasticity via Pretraining with Active Forgetting. NeurIPS 2023.

[4] Pfeiffer et al. MAD-X: An Adapter-Based Framework for Multi-Task Cross-Lingual Transfer. EMNLP 2020.

[5] Pfeiffer et al. Lifting the Curse of Multilinguality by Pre-Training Modular Transformers. NAACL 2022.

---

> ### Author Rebuttal · Authors · 2026-03-31
>
> We thank the reviewer for recognizing the intuition and novelty of our Markov modeling perspective. Below, we address the comments sequentially.
>
> **Response to W1:** We would like to clarify that ''knowledge expansion'' is a broader concept, while ''vocabulary expansion'' in our work is a specific mechanism used to achieve it. In particular, we introduce new tokens as interfaces to encode additional knowledge, but the key contribution lies in how these tokens are mapped into the existing representation/transition structure, rather than the increase in vocabulary itself. We will make this scope clearer in the revision.
>
> **Response to W2/Q1:**
> Although our datasets are synthetic, they are constructed from real language sentences. This design provides a controlled setting to evaluate the effectiveness of our embedding tuning method and to validate our theoretical findings, where we assume a *small* number of new tokens ($m\ll T$) and that they are *independent*.
>
> We appreciate the reviewer’s suggestion regarding real vocabulary expansion. In our paper, we do not treat this as knowledge expansion, since modern LLMs already cover most languages and typically do not require adding new tokens (This is also why we adopt a synthetic setup to precisely match the assumptions of our theoretical analysis, which makes sure the new tokens have never been seen before). Regardless of the theoretical assumptions, we conducted additional experiments following the reviewer’s suggestion. Interestingly, we observe similar results! Experimental details are provided below.
>
> We use Qwen2.5-3B as the backbone model and build datasets from Wikipedia, including 1000 Spanish training sentences, 1000 Spanish test sentences, and 1000 English test sentences (to test forgetting). Additionally, we mine 500 frequent Spanish words that are not atomic tokens in the base tokenizer, and add them as new vocabulary entries. Experimental results are presented below.
>
> | Method | Perf. (Spanish) | Forgetting (English) |
> |----------|-----------------|--------------------------|
> FFT | 5.56 | +9.83|
> LoRA | 3.17 | +0.66|
> PT |3.45 | -0.0315|
> ET | **2.30**| **-0.0352**|
>
> The same results hold that ET performs best among all methods. Surprisingly, the forgetting can be even negative. One interpretation is that: adding and tuning new tokens can refine the structure of the embedding space (e.g., improving alignment), which may also benefit English performance. This is an interesting observation and opens a promising direction for future work. We will include additional experiments (e.g., more datasets and models) and expand the discussion in the final revision.
>
> **Response to W3:**
> We thank the reviewer for highlighting this relevant literature. We will add a paragraph on the related work to compare our framework more clearly with prior methods based on embedding relearning/reinitialization and modular transfer. Our contribution is complementary to these approaches: we focus on sample-efficient vocabulary expansion in autoregressive LMs, with theoretical guarantees under a Markov formulation.
>
> **Response to Q2:**
> We believe our theory is related in spirit but not directly applicable as stated. The closest relevant work is Chen et al. (2023) [1], which proposes periodically resetting the embedding layer every $K$ updates during **pretraining** so that the model can adapt to new languages faster, especially in low-data settings.
>
> In contrast, our theory studies a different regime: a **post-pretraining** vocabulary expansion setting in which the pretrained transition structure on the original vocabulary is preserved, and only the representations of newly introduced tokens are learned through a token-to-dictionary mapping / embedding-tuning mechanism. Under this setup, our theoretical and empirical guarantees explain why learning new tokens can be sample-efficient and why the original vocabulary can be preserved without forgetting.
>
> [1] Chen et al. ''Improving language plasticity via pretraining with active forgetting.'' NeurIPS 2023.
>
> **Response to Q3:**
> Under our theory, the preservation claim is only about the original pretrained vocabulary: the transition dynamics among original tokens are kept unchanged, while the newly added tokens learn their own interactions with that fixed structure. Thus, ET does not preserve all token-token relationships globally; it preserves the old-old relationships while learning new-old ones. This guarantee relies on updating only the embeddings of newly added tokens. If one instead updates a subset of existing pretrained token embeddings, then relationships between that subset and the remaining pretrained tokens can indeed change, and the same no-forgetting guarantee would no longer apply.

---

> > ### Author Rebuttal · Reviewer_UvGz · 2026-04-03
> >
> > The questions are elaborated in my comments. Sorry the comment above is not visible to the author for some reason. I pasted it below.
> >
> > Thank you for the rebuttal.
> >
> > **On W1 (knowledge expansion vs vocabulary expansion):**
> > Thanks for clarification. The difference between knowledge expansion (as the broader objective) and vocabulary expansion (as the mechanism) is now clearer. I encourage the authors to make this *explicit* early in the paper to avoid confusion about the scope of the contribution, especially since works such as [1]–[5] are typically viewed as vocabulary expansion in the NLP literature.
> >
> >
> > **On W2 / Q1 (synthetic setup and practical relevance):**
> > I appreciate the additional Spanish experiment. This highlights the potential of using your method to increase the multlinguality of LLMs. The observation of negative forgetting is particularly interesting but there can be other factors explaining the negative forgetting -- it could be related to how Qwen2.5-3B was trained (e.g. they were poorly trained or distilled from larger models) or the limited evaluation set of 1000 English sentences. In general, it will be more convincing if you can extend these experiments to more backbone models and larger evaluation set. Additionally, the language distance between Spanish and English are small compared to between Thai and English [3]. The real world challenge usaually are between the more distant language pairs. I would expect your methods to be more useful in these distant cases
> >
> > I remain unconvinced by the claim that real vocabulary expansion is not practically relevant. In realistic settings, new vocabulary (e.g., domain-specific terminology, emerging concepts, or named entities) is continuously introduced into each language in the world. Efficiently incorporating such tokens without full retraining is an important use case. In this regard, I view the additional experiment as supporting the *practical importance* of the problem and necessary to include in the final version. I encourage the authors to revise the paper to better reflect this.
> >
> > **On W3 (related work):**
> > Thank you for acknowledging the missing literature. Incorporating and clearly positioning the work relative to cross-lingual transfer and vocabulary expansion [1] -[5] (e.g., embedding relearning, modular transfer) will significantly strengthen the paper. In particular, it would be helpful to more clearly articulate what is novel in the proposed Markov formulation compared to these prior approaches and how researchers from downstream communities e.g. NLP, can benefit from your method.
> >
> > **On Q2 (connection to embedding resetting):**
> > I agree that the connection is indirect. A brief discussion situating your framework relative to works such as Chen et al. (2023) would improve the paper’s positioning within the broader literature on plasticity and continual learning. For example, your methods are functioning at a different phase from [3] (which require pretraining intervention) but addressing the same issue of plasticity in continual learning.
> >
> > **On Q3 (preservation of token relationships):**
> > The clarification is helpful.
> >
> > **Overall:**
> > The rebuttal addresses several of my concerns, and the additional experiment is a positive signal. If the authors incorporate the above clarifications and claimed revisions, particularly on improving problem framing, discussion of practical relevance, and better positioning relative to prior work, I will increase my score.
> >
> > ---
> >
> > April 7th: Thank you for responding to my rebuttal comments and for reporting the additional En–German results. I would not expect En–German to differ substantially from En–Spanish. My intuition is that your method is likely to be more useful for more distant language pairs, which would help bridge the gap between theory and realistic settings highlighted in Reviewer 8aW5’s evaluation. If you can provide experiments along these lines, it would demonstrate the practicality and real-world relevance of your approach for other reviewers and general readers. For now, I will raise my confidence score and significance score for this submission.

---

> > > ### Author Response · Authors · 2026-04-03
> > >
> > > ----
> > > **[Original response on Apr 3]**
> > >
> > > We thank the reviewer for the update. Since the follow-up questions are not specified in the comment, we would greatly appreciate it if the reviewer could clarify the remaining concerns so that we can address them more precisely.
> > >
> > > In the meantime, we would like to briefly restate our main contributions: As noted by the reviewer, our work establishes **a principled connection between LLMs and Markov chains, supported by both theoretical analysis and empirical validation**. As a first step, we focus on the setting of knowledge expansion, which enables a clean and controlled study of the problem. We believe this formulation provides a novel perspective for research on LM output distributions.
> > >
> > > Regarding the reviewer’s main practical concern, we have incorporated the additional experimental setting suggested by the reviewer during the rebuttal. The results further support the effectiveness of our method and are consistent with our theoretical insights.
> > >
> > > We would be happy to provide further clarification once the remaining questions are specified.
> > >
> > > --------
> > > **[Updated response on Apr 6]**
> > >
> > > We thank the reviewer for the constructive follow-up and positive signal.
> > >
> > > **On W1 (framing):** We will add a clear paragraph to the paper explicitly distinguishing knowledge expansion (the goal) from vocabulary expansion (the mechanism), and contrast this with the NLP literature's use of the term.
> > >
> > > **On W2/Q1 (synthetic setup and practical relevance):** We would like to thank the reviewer again for the suggestion on the additional experiments! Indeed, we are currently running further experiments (using different backbone models and translation tasks). During the rebuttal phase, we also tried English-to-German. However, due to the character limit and the results that are consistent with the English-Spanish setting, we did not include them in our original response. The results are
> > >
> > > | Method|Loss (German)|Forget (English)|
> > > |---|---|---|
> > > |FFT| 6.80 | +9.81|
> > > |LoRA| 3.29 | +0.49|
> > > |PT| 3.63 | -0.028|
> > > |ET| **3.27** | **-0.078**|
> > >
> > > Motivated by the reviewer's additional comment, we will include a more distant language pair (e.g., Thai or Arabic) to stress-test the method under larger vocabulary gaps.
> > >
> > > Regarding the negative forgetting, we agree that multiple factors may contribute. Due to the complexity and limited interpretability of LLMs, it is difficult to provide a definitive explanation at this stage. We will include a more thorough and careful analysis in the revised manuscript as additional experimental results are obtained.
> > >
> > >
> > >
> > > Our "*real vocabulary expansion is not practically relevant*" statement was based on the assumption that current LLMs already have broad language coverage. However, the "*new vocabulary (e.g., domain-specific terminology, emerging concepts, or named entities) is continuously introduced*" scenarios as highlighted by the reviewer, are highly relevant and directly aligned with the problem our method aims to address. We will reframe our practical motivation to include those cases as valid use cases.
> > >
> > > **On W3/Q2/Q3:** We have been working on revising our submitted manuscript and commit to incorporating these discussions.
> > >
> > > **Overall:** We appreciate the reviewer’s additional thoughtful comments and hope our responses address all concerns. As the response deadline approaches, we will continue to monitor any further comments from the reviewer and update our response promptly if necessary.
> > >
> > > -------
> > >
> > > **[Final update on Apr 7]**
> > >
> > > We thank the reviewer for the suggestion on distant language pairs. Following this, we completed **English-to-Arabic** experiments across three backbone models. Arabic is a significantly more distant language pair than Spanish or German, directly addressing the reviewer's concern.
> > >
> > > | Model | Arabic Loss (FFT / LoRA / PT / ET) | English Forgetting (FFT / LoRA / PT / ET) |
> > > |---|---|---|
> > > | Qwen2.5-3B  | 7.95 / 3.95 / 4.34 / **2.82** | +11.95 / +0.46 / +0.06 / **0.00** |
> > > | Llama3.2-1B  | 5.27 / 3.65 / 6.05 / **3.47** | +13.59 / +0.35 / +0.42 / **+0.0001** |
> > > | Llama3.2-3B  | 7.43 / 3.57 / 5.62 / **3.15** | +14.22 / +0.39 / +0.37 / **+0.0003** |
> > > **Key observations:**
> > > 1. **ET** achieves the best *Arabic* performance across **all three models**, confirming the results are **not model-specific or limited to closely related language pairs**.
> > > ---
> > > 2. **ET** achieves **near-zero English forgetting (0.00, 0.0001, 0.0003)** consistently across all models, ruling out *Qwen2.5-3B* artifact. The *Llama* results confirm this is a **robust property** of the method.
> > > ---
> > > 3. The **ET**-baseline gap is larger for Arabic than Spanish/German, consistent with the reviewer's intuition and directly addressing Reviewer **8aW5**'s theory-experiment gap concern.
> > > ---
> > > We will incorporate these results and all committed revisions into the final manuscript. We would greatly appreciate a reconsideration of the overall score.

---

### Official Review · Reviewer_8aW5 · 2026-03-16

**Soundness:** 2
**Presentation:** 2
**Significance:** 2
**Originality:** 2
**Overall Recommendation:** 3
**Confidence:** 3

**Summary:**

This paper proposes a framework for adding a small amount of new knowledge to an LLM without catastrophic forgetting by viewing next-token generation as a Markov process over tokens, where learning new knowledge amounts to expanding the state space while preserving old transition dynamics. The key underlying mechanism is a token-to-dictionary mapping, where each new token is represented using a sparse combination of existing tokens, which leads to a sample-complexity result. The authors show that in controlled experiments, this can learn the new behavior while maintaining zero forgetting on previous knowledge.

**Compliance With Llm Reviewing Policy:**

Affirmed.

**Key Questions For Authors:**

How does this paper differ from Nested Learning ( Nested Learning: The Illusion of Deep Learning Architectures, Neurips 2025), which is also motivated by mitigating catastrophic forgetting? A brief discussion of the conceptual connection might be useful.

**Limitations:**

yes

**Strengths And Weaknesses:**

**Strengths**
1) Unlike regularization methods (e.g., EWC) or PEFT methods like LoRA that only mitigate forgetting, the ET approach ensures zero degradation of the original vocabulary because the transition computation graph for those tokens is never modified.

2) The method is extremely parameter-efficient, updating only a few thousand parameters compared to millions in LoRA.

**Weaknesses**

1) The formal no-forgetting story relies on two strong assumptions: no transitions from old tokens to new tokens, and no transitions among new tokens. This means the theory does not cover the more natural case where the model should learn to spontaneously produce the newly introduced token in ordinary contexts, or where newly added tokens interact with each other.

2) The authors acknowledge this and discuss a higher-order extension, but that section is much more heuristic than the earlier theory. In particular, the move from exact first-order results to arguments based on “meaningful states” and effective branching factor feels plausible but not nearly as rigorous as the main theorems.

3) In the synthetic benchmark, the authors note that the synthetic data is generated using Phi-3.5 Mini Instruct itself, which may favor prompt tuning on that model. That is a fair acknowledgment, but it also suggests the evaluation setup may be partly entangled with the generator model. More discussion would help.


**Typose**
1) aims to identifying  should be aims to identify (line 175-176)

---

> ### Author Rebuttal · Authors · 2026-03-31
>
> We thank the reviewer for the thoughtful feedback. We addressed the concerns raised by the reviewer below.
>
> **Response to W1:**
> First we would like to clarify that our experiments are less restrictive than the theory. In our experiments (Sec. 5.2), we train with standard *next-token prediction* on language sentences. So, in practice, old-to-new and new-to-new transitions do occur. In our response to the **Reviewer UvGZ's, W2**, we discuss additional experiments where new tokens are denser and interact with each other.
>
> Regarding our theoretical assumptions:
> 1. **Old to new transitions:** We assume no transitions from old to new tokens to enable a clean quantification of model forgetting. Without this assumption, measuring forgetting becomes more challenging. Regardless of the forgetting analysis, our sample complexity analysis can be extended to allow such transitions. In this case, similar theoretical guarantees hold, requiring $O(s \log^2c)$ samples per token, where $c$ denotes the lower bound on the **ingress (incoming)** transition probabilities. In our paper, we focus on **next-token (egress, outgoing)** probabilities in the autoregressive setting.
> 2. **New to new transitions:** We emphasize that allowing transitions among new tokens does not introduce any forgetting. Our current theory assumes no such transitions for simplicity, corresponding to a sparse regime where $m\ll T$. However, when interactions among new tokens exist, the analysis can be extended: informally, the sample complexity scales as $O(m\cdot \log^2 c)$ with $c$ depending on the transition structure within the $m\times m$ submatrix.
>
> A more precise analysis would require additional assumptions on the model formulation and transition structure. We will add this discussion to the manuscript.
>
> **Response to W2:**
> We did not expand the higher-order Markov extension in detail because, as discussed in Sec. 4.3, the core theoretical results extend directly when considering an enlarged state space. Specifically, for a $K$-order Markov model, one can reformulate it as a first-order Markov process over an expanded state space of size $Tb^{K-1}$, where $b$ is the effective branching factor. Under this formulation, Theorems 4.5 and 4.7 apply directly (by replacing $T$ with  $Tb^{K-1}$).
>
> The key challenge then lies in identifying a set of semantically meaningful states $\bar{\mathcal{V}}$, that is, determining an appropriate factor $b$. However, it is beyond the scope of this work, where we focus on model training rather than analysis of linguistic structure. We will enrich the discussion on this point in the revision.
>
> **Response to W3:** The paper notes that the synthetic benchmark is generated using Phi-3.5 Mini Instruct, which may help explain why prompt tuning performs particularly well on Phi in this setting. One plausible explanation is that, because the synthetic data is produced by the same model family, the adaptation problem may be well matched to the model’s existing representations and prompting behavior, which can favor prompt tuning.
>
> In contrast, real-world data is typically more natural and human-authored, and thus less likely to exhibit this same kind of alignment. We therefore agree that the strong PT performance observed here may be partly specific to this synthetic setting and should not be overgeneralized. At the same time, the precise mechanism behind this effect remains unclear, and we can only provide empirical observations and hypotheses at this stage. We will expand the discussion in the revision and view this as an interesting direction for future work, particularly for developing a more rigorous theoretical understanding.
>
> **Response to Q:**
> There is a meaningful conceptual connection between our work and Nested Learning: both are motivated by the continual-learning setting and by the challenge of mitigating catastrophic forgetting. However, the two works operate at very different levels.
> - Nested Learning is a learning paradigm that models a system as a collection of nested, multi-level optimization problems and develops new optimizer/memory/architecture ideas. Its emphasis is therefore on architectural and optimization redesign for continual learning.
> - In contrast, our paper studies a more formalized question: how to incorporate new knowledge into a pretrained LM without forgetting, by modeling next-token generation as a Markov process, viewing new-token addition as state-space expansion, proving sample-complexity guarantees for token-to-dictionary mapping, and realizing this with embedding tuning that preserves prior behavior.
>
> Thus, we see the two works as complementary rather than competing. We will add a brief discussion to make this distinction explicit.

---

> > ### Author Rebuttal · Reviewer_8aW5 · 2026-04-04
> >
> > I thank the authors for the response. I appreciate the clarifications, but I am not changing my score because the main issues were not fully resolved within the current paper. In particular, the rebuttal mainly argues that the theory can be extended, but those extensions are not actually developed or validated in the submission, so the gap between the clean theoretical guarantees and the more realistic experimental setting still remains.

---

> > > ### Author Response · Authors · 2026-04-05
> > >
> > > Thank you for the follow-up. Based on the reviewer’s additional comment, we believe our rebuttal has addressed the main concerns. Since we are not allowed to modify the manuscript during the rebuttal phase, these clarifications could not be incorporated into the main paper.
> > >
> > > We would like to further emphasize the following points:
> > >
> > > 1. **Our theoretical analysis can be directly extended.** The extensions to other transition patterns (old-to-new and new-to-new) *do not require fundamentally new techniques*. They follow from the same analytical framework with mild modifications, by adjusting Assumption 4.2 (egress probability), Lemma A.1 (probability lower bound) and applying standard Markov chain concentration arguments (spectral gap of the new-token sub-chain). *We chose to state the theory in the clean restricted regime to keep the presentation focused, not because the general case requires fundamentally different proof techniques.* We will make this distinction explicit in the revision.
> > >
> > > 2. **Our experiments do not impose these restrictions.** The experiments in Sec. 5.2, as well as the English-to-Spanish expansion experiment added during the rebuttal, directly evaluate the method *in an unrestricted and more practical setting.* These results demonstrate that the method remains effective beyond the simplified theoretical assumptions, thereby providing empirical support that complements the theory.
> > >
> > > 3. **Why do we focus on new-to-old transitions in the paper?** The theoretical formulation omits old-to-new transitions primarily *to enable a precise and interpretable notion of forgetting*. Otherwise, there is no clear and formal way to quantify forgetting. Importantly, if the objective is sample efficiency alone (independent of forgetting), our theoretical results extend directly to these broader settings. Similarly, we place less emphasis on new-to-new transitions since the newly introduced token set $\mathcal{U}$ is typically small and sparse relative to the original vocabulary $\mathcal{V}$. That said, we agree with the reviewer that explicitly discussing these extensions would strengthen the paper, and we will incorporate this discussion in the revision.
> > >
> > > Overall, we believe that the combination of a novel and principled theoretical framework (which to the best of our knowledge has not been explored in prior work) together with supporting empirical results, constitutes a complete and meaningful contribution. We would greatly appreciate your reconsideration of the score.
> > >
> > >
> > > **[update on Apr 7]**
> > >
> > > We would like to draw the reviewer's attention to our latest update in Reviewer **UvGz**'s thread, where we have added **English-to-Arabic** experiments across three backbone models *(Qwen2.5-3B, Llama3.2-1B, Llama3.2-3B)*. **ET** consistently achieves the **best performance and near-zero forgetting** on a genuinely distant and practical language pair, providing empirical support that directly addresses the **theory-experiment gap concern**. We would greatly appreciate a reconsideration of the score.

---

### Decision · Program_Chairs · 2026-04-30

**Decision:**

Accept (regular)

**Comment:**

This paper studies how to add new knowledge to a pretrained language model without retraining it or degrading what it already knows. The authors propose to treat the model’s next-token predictions as a Markov process over tokens, and to incorporate new concepts by introducing new tokens that are expressed as combinations of existing ones, learned through a lightweight embedding update.

The main strength of the paper lies in this modeling choice: interpreting LLMs as a Markov transition matrix and casting knowledge expansion as state-space expansion provides a clear and original lens on continual learning. The accompanying sample complexity analysis gives a principled account of why embedding-only updates can suffice, and was generally seen as technically sound. While the initial experimental section relied on controlled synthetic tasks, the additional results provided during rebuttal, particularly on cross-lingual transfer, including distant language pairs, offer encouraging evidence that the method can extend beyond idealized settings and retain its no-forgetting behavior in practice.

At the same time, reviewers raised some important concerns. The most significant is the gap between theory and practice: the main guarantees rely on restrictive assumptions (e.g., no transitions from old to new tokens or among new tokens), which do not hold in realistic generation scenarios. Although the authors argue that these assumptions can be relaxed, these extensions are not fully developed or rigorously validated in the current version. Relatedly, the empirical evaluation in the paper remains somewhat limited in scope, with the more compelling experiments only appearing in the rebuttal and not yet integrated into a cohesive experimental section. Finally, the positioning could be clearer, particularly in distinguishing “knowledge expansion” from vocabulary expansion and situating the work more precisely with respect to prior literature on cross-lingual transfer and embedding-based adaptation.

After considering the reviews, author responses, and discussion, I believe the paper should be accepted. Its central contribution, a novel and well-motivated Markovian formulation of LLM memory with non-trivial sample complexity results, is technically sound and provides useful insight into embedding-based adaptation, making it relevant to the ICML community. While the theoretical assumptions are simplified, the combination of analysis and supporting empirical evidence suggests the framework captures meaningful aspects of practical behavior. The rebuttal further strengthened the submission by clarifying assumptions, addressing key concerns, and adding additional experiments.

For the camera-ready version, the authors should incorporate all edits from discussion, in particular:
- more clearly delineate which theoretical results hold under which assumptions and discuss extensions beyond the restrictive setting.
- integrate the new cross-lingual experiments with sufficient detail and analysis.
- improve positioning with respect to prior work and clarify the scope of knowledge expansion.
- explicitly state the limitations regarding spontaneous generation and applicability beyond token addition.